# FLOW SCORE DISTILLATION FOR DIVERSE TEXT-TO-3D GENERATION

## ABSTRACT

Recent advancements in Text-to-3D generation have yielded remarkable progress, particularly through methods that rely on Score Distillation Sampling (SDS). While SDS exhibits the capability to create impressive 3D assets, it is hindered by its inherent maximum-likelihood-seeking essence, resulting in limited diversity in generation outcomes. In this paper, we discover that the Denoise Diffusion Implicit Models (DDIM) generation process (i.e. PF-ODE) can be succinctly expressed using an analogue of SDS loss. One step further, one can see SDS as a generalized DDIM generation process. Following this insight, we show that the noise sampling strategy in the noise addition stage significantly restricts the diversity of generation results. To address this limitation, we present an innovative noise sampling approach and introduce a novel text-to-3D method called Flow Score Distillation (FSD). Our validation experiments across various text-to-image Diffusion Models demonstrate that FSD substantially enhances generation diversity and quality. Project page: `https://flowscoredistillation.github.io/`

## 1 INTRODUCTION

In the realm of 3D content creation, a crucial step within the modern game and media industry involves crafting intricate 3D assets. Recently, 3D generation has facilitated the creation of 3D assets with ease. 3D generative models could be trained directly on certain representations (e.g. point clouds (Achlioptas et al., 2018; Luo & Hu, 2021), voxel (Xie et al., 2018; Smith & Meger, 2017) and mesh (Zhang et al., 2021)). However, despite the recent efforts of Objaverse (Deitke et al., 2023), 3D data remains relatively scarce, especially when compared to the abundant 2D image data available on the internet. This scarcity constrains the generative capabilities of models trained solely on 3D datasets. Notably, the most prevailing text-to-3D approach is based on Score Distillation Sampling (SDS), proposed by Dreamfusion (Poole et al., 2022) and SJC (Wang et al., 2023a). SDS effectively tackles the scarcity of 3D data by leveraging pretrained 2D text-to-image Diffusion Models, without directly training models on 3D datasets.

SDS is designed to optimize any representations (e.g. Neural Radiance Field (Mildenhall et al., 2021; Müller et al., 2022; Wang et al., 2021), 3D Gaussian Splatting (Kerbl et al., 2023), Mesh (Laine et al., 2020; Shen et al., 2021) or even 2D images) that could render 2D images through probability density distillation (Oord et al., 2018) using the learned score functions from the Diffusion Models. One of the main limitations of current SDS-based methods is that their distillation objectives will maximize the likelihood of the image rendered from the 3D representations, which leads to limited diversity. Despite several subsequent efforts (Wang et al., 2024; Zhu & Zhuang, 2023; Liang et al., 2023; Katzir et al., 2023; Huang et al., 2023; Tang et al., 2023; Wang et al., 2023b; Armandpour et al., 2023) to enhance SDS, the maximum-likelihood-seeking essence of the method remains unchanged. Notably, ProlificDreamer (Wang et al., 2024) introduces Variational Score Distillation (VSD) and uses a fine-tuned Diffusion Model to model distribution on particles, which could alleviate the maximum-likelihood-seeking issues. However, training costs could grow linearly with the particle number of VSD. ESD (Wang et al., 2023b) points out that single-particle VSD is equivalent to SDS, which remains rooted in the essence of maximum likelihood seeking.

In this paper, we present a fresh perspective on SDS by viewing it as a generalized DDIM (Song et al., 2020a) generation process for 3D representations. Specifically, we discovered that the DDIM

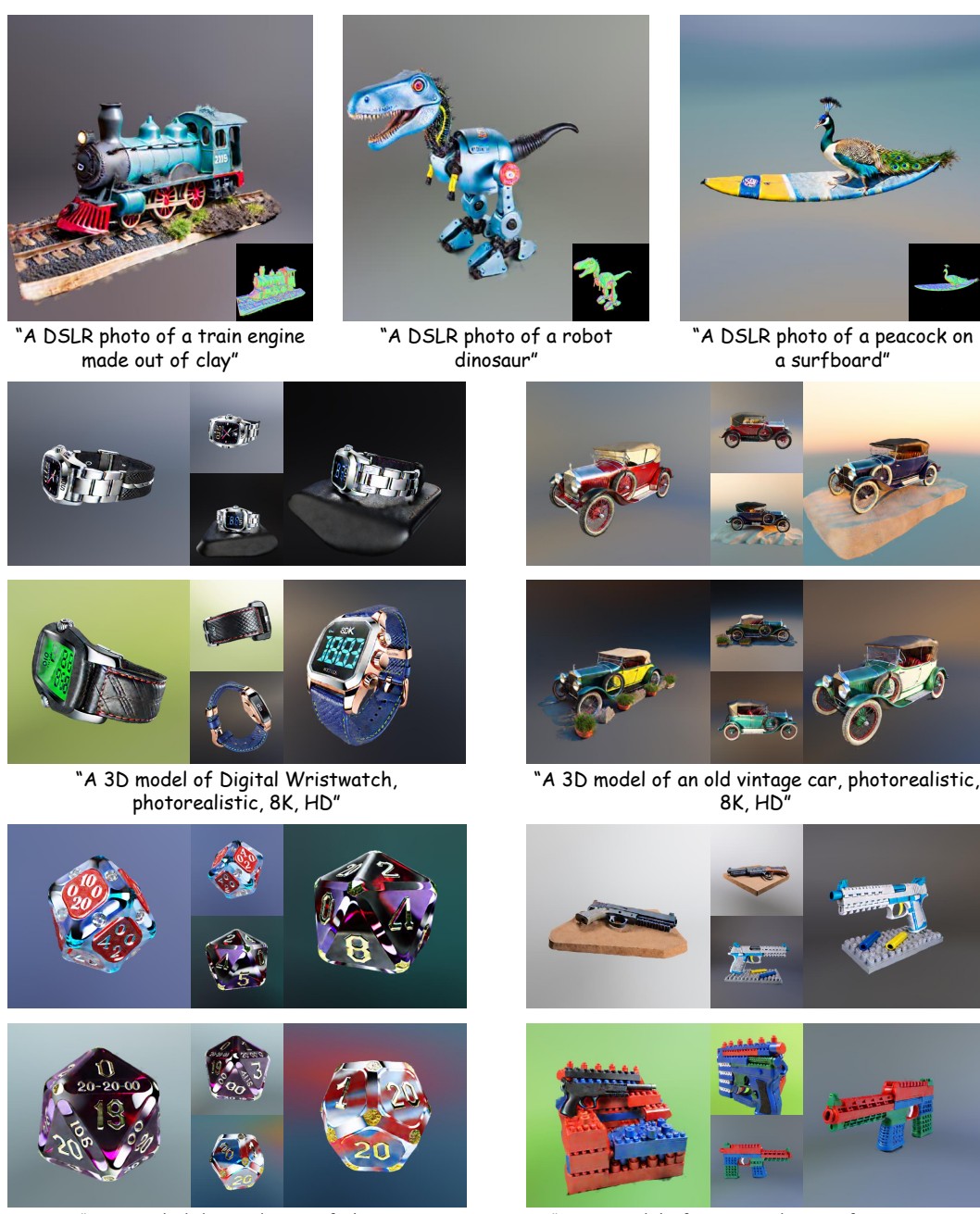

Figure 1: **Generation results of FSD and baseline method SDS. FSD** uses pretrained text-to-image Diffusion Models to generate realistic 3D models from text prompts. We improve the noise sampling strategy upon SDS and achieve diverse generation results with high quality.

generation process (i.e. PF-ODE (Song et al., 2020b)) can be succinctly expressed using an analogue of SDS loss. Surprisingly, by studying the difference between the analogue of SDS loss and the original form of SDS loss, we find that the noise sampling strategy during the noise addition stage appears to be the main cause that drives SDS toward mode-seeking behavior. SDS-based methods typically use random noise sampled from a Gaussian distribution at each optimization step, following the proposal of Dreamfusion (Poole et al., 2022) and SJC (Song et al., 2020b). However, the variation in sampled noise can lead to varied optimization directions, which may harm the perfor-

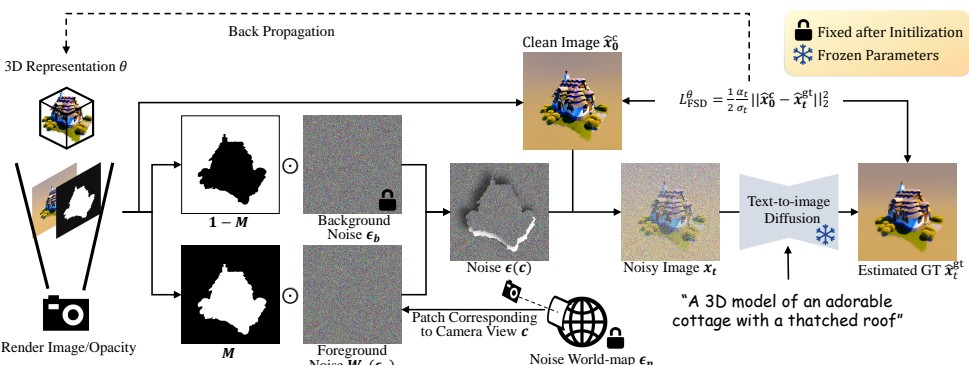

Figure 2: **Methods overview of FSD.** We propose Flow Score Distillation for text-to-3D generation by lifting a pretrained Diffusion Model. FSD renders an image $g_\theta(c)$ from the 3D representation and adds noise $\epsilon(c)$ to the rendered image. To compute parameter updates according to $L^\theta_{\text{FSD}}$, FSD uses a frozen text-to-image Diffusion Model to predict the noise $\epsilon(c)$ added on image $g_\theta(c)$. Similar to SDS (Poole et al., 2022; Wang et al., 2023a), FSD computes $L^\theta_{\text{FSD}}$ by an image reconstruction loss between the "clean image" $\hat{x}^c_t = g_\theta(c)$ and "ground-truth image" $\hat{x}_0$ predicted by the pretrained Diffusion Model. FSD further adopts timestep annealing schedule and noise sampling strategy. Instead of sampling noise from Gaussian distribution at each step of the optimization like SDS, we generate noise according to the deterministic noise function $\epsilon(c)$, which is determined at the beginning of the optimization.

mance of SDS, as observed by ISM (Liang et al., 2023). As we will show in this work, PF-ODE can be expressed by an analogue of SDS loss that uses a fixed noise throughout the generation process, rather than randomly sampled noise, which is different from the original proposal of SDS (Poole et al., 2022; Wang et al., 2023a). Based on this insight, we propose a novel noise sampling strategy to align SDS with the DDIM generation process on 3D representations.

This paper aims to overcome the aforementioned diversity challenge of SDS. We will first reveal an underlying connection between SDS and DDIM on 2D image generation. We will also show that the noise sampling strategy could be the primary factor that leads to the restricted diversity. Based on this insight, we will give our interpretation of SDS. From our novel viewpoint on SDS, we propose a novel approach called *Flow Score Distillation* (FSD). FSD improves SDS by using a carefully designed noise sampling strategy. We lift our observations on image generation with FSD to 3D by proposing a view-dependent noise function $\epsilon(c)$. We conduct validation experiments across various 2D Diffusion Models and demonstrate that FSD can achieve diverse generation outcomes with high quality while introducing no extra training costs. Finally, we propose a 2 stage coarse to fine generation pipeline for high quality text-to-3D generation to overcome multi-face problems. We use pretrained text-to-muiltview-image diffusion model to generate the coarse shape and then refine the details using pretrained text-to-image diffusion model. The generation results are presented in Fig. 1. Overall, our contributions can be summarized as follows.

- We provide an in-depth analysis of SDS, an effective method in text-to-3D generation. Specifically, the DDIM generation process (i.e. PF-ODE) can be succinctly expressed using an analogue of SDS loss. As a result, we can interpret SDS as a generalized DDIM generation process on 3D representations where a fixed noise is added.

- Building upon our new insight into SDS, we introduce FSD as a cheap but effective solution to tackle the diversity challenges arising from the maximum-likelihood-seeking nature of SDS. We propose a deterministic *world-map noise function* to generate coarsely aligned noise in 3D space. By applying a reasonable noise sampling strategy, FSD breaks free from the mode-seeking nature of SDS-like methods.

- We propose a 2 stage coarse to fine pipeline to tackle multi-face problems. By incorporating a multiview diffusion model that has rich shape prior in coarse stage and a image diffusion model that has rich texture prior in refine stage, our methods can generate diverse and high quality 3D objects.

## 2 PRELIMINARIES AND RELATED WORKS

### 2.1 DIFFUSION MODELS

Diffusion Models (Sohl-Dickstein et al., 2015; Ho et al., 2020; Song et al., 2020b) are a family of powerful generative models that are trained to gradually transform Gaussian noise to samples from a target distribution $p_0$. Their generation ability is further enhanced given datasets comprising billions of image-text pairs (Changpinyo et al., 2021; Schuhmann et al., 2022; Sharma et al., 2018).

Assume the target distribution is $p_0$ and condition $y$. Diffusion Models define a forward process $\{x_t\}_{t \in [0,T]}$ starting from $\boldsymbol{x}_0 \sim p_0(\cdot|y)$, such that for any $t \in [0,T]$ the distribution of $x_t$ conditioned on $x_0$ satisfies:

$$\boldsymbol{x}_t = \alpha_t \boldsymbol{x}_0 + \sigma_t \boldsymbol{\epsilon}, \quad \boldsymbol{\epsilon} \sim \mathcal{N}(\mathbf{0}, \boldsymbol{I}), \quad \boldsymbol{x}_0 \sim p_0(\cdot|y), \tag{1}$$

where $\alpha_t, \sigma_t \in \mathbb{R}^+$ are functions of $t$, defined by the *noise schedule* of the Diffusion Model. And the (noisy) distribution at timestep $t$ is noted as $p_t$.

In practice, a Diffusion Model is a neural network $\boldsymbol{\epsilon}_\phi(\boldsymbol{x}_t|y,t)$ parameterized by $\phi$ and is trained by minimizing the following score matching objective (Song et al., 2020a;b):

$$L_{\text{DMs}}^\phi = \frac{1}{2} \mathbb{E}_{\boldsymbol{x}_0 \sim p_0(\cdot|y), \boldsymbol{\epsilon}, t} \left[ w_t || \boldsymbol{\epsilon}_\phi(\boldsymbol{x}_t|y,t) - \boldsymbol{\epsilon}||_2^2 \right], \tag{2}$$

where $w_t$ is a weighting function. Song et al. (2020b) proved that:

$$\boldsymbol{\epsilon}_\phi(\boldsymbol{x}_t|y,t) \approx -\sigma_t \nabla_{\boldsymbol{x}} \log p_t(\boldsymbol{x}_t|y), \tag{3}$$

if the Diffusion Model $\boldsymbol{\epsilon}_\phi$ is trained to almost optimum. And the term $\nabla_{\boldsymbol{x}} \log p_t(\boldsymbol{x}_t|y)$ in the above equation is also known as the *score function*.

### 2.2 DIFFUSION PF-ODE AND DDIM

To generate samples from Diffusion Models, there exist different methods among the Diffusion Models family. Denoise Diffusion Implicit Models (DDIM) (Song et al., 2020a) designed a deterministic method for fast sampling from Diffusion Models. Later works (Salimans & Ho, 2022; Karras et al., 2022; Lu et al., 2022) showed that the sampling algorithm of DDIM is a first-order discretization of the Probability Flow Ordinary Differential Equation (PF-ODE) (Song et al., 2020b).

Theoretically, Diffusion PF-ODE yields the same marginal distribution as the forward process of Diffusion Models (Eq. 1) (Song et al., 2020b). We can write Diffusion PF-ODE (Karras et al., 2022; Song et al., 2020b) (see detailed derivation in Appx. Sec. F.1) as:

$$\frac{\mathrm{d}(\boldsymbol{x}_t/\alpha_t)}{\mathrm{d}t} = \frac{\mathrm{d}(\sigma_t/\alpha_t)}{\mathrm{d}t} \left( -\sigma_t \nabla_{\boldsymbol{x}} \log p_t(\boldsymbol{x}_t|y) \right) \tag{4}$$

$$= \frac{\mathrm{d}(\sigma_t/\alpha_t)}{\mathrm{d}t} \boldsymbol{\epsilon}_\phi(\boldsymbol{x}_t|y,t), \quad \boldsymbol{x}_T \sim p_T(\boldsymbol{x}_T|y). \tag{5}$$

Notably, one can generate a sample $\boldsymbol{x}_0$ in the target distribution $p_0(\boldsymbol{x}_0|y)$ by following the PF-ODE trajectory from $t = T$ to $t = 0$, starting from $\boldsymbol{x}_T \sim p_T(\boldsymbol{x}_T|y) = \mathcal{N}(\mathbf{0}, \boldsymbol{I})$.

### 2.3 SCORE DISTILLATION SAMPLING

Recently, DreamFusion (Poole et al., 2022) and SJC (Wang et al., 2023a) proposed Score Distillation Sampling (SDS) to generate 3D models by optimizing a differentiable 3D representation using priors from text-to-image Diffusion Models. Follow-up works tried to improve upon SDS through various aspects, e.g. coarse-to-fine training strategy (Lin et al., 2023; Wang et al., 2024; Chen et al., 2023), disentangled 2D-3D priors (Chen et al., 2023; Ma et al., 2023; Wang et al., 2024) and refined formulas (Zhu & Zhuang, 2023; Wang et al., 2024; Liang et al., 2023; Tang et al., 2023; Wang et al., 2023b; Yu et al., 2023; Armandpour et al., 2023). Moreover, due to the lack of comprehensive 3D-aware knowledge, multi-face Janus problem often arises when using SDS (Poole et al., 2022). To mitigate this challenge, one can consider replacing the text-to-image Diffusion Models with Diffusion Models designed for object novel view synthesis (Liu et al., 2023b; Long et al., 2023; Liu et al., 2023c; Weng et al., 2023; Ye et al., 2023) or multi-view Diffusion Models (Shi et al., 2023). Such an adaptation can alleviate the multi-face Janus problem encountered in 3D generation using SDS.

SDS is first introduced by DreamFusion (Poole et al., 2022) and SJC (Wang et al., 2023a) to apply image diffusion priors for 3D generation. SDS can optimize on any representations parameterized by $\theta$, which can render an image $\boldsymbol{g}_\theta(\boldsymbol{c})$, given camera parameter $\boldsymbol{c}$. Basically, SDS defines a probability density distillation (Oord et al., 2018) loss, denoted as $L_{\text{SDS}}^\theta$, whose gradient writes as follows:

$$\nabla_\theta L_{\text{SDS}}^\theta = \mathbb{E}_{\boldsymbol{c},t}\left[w_t \frac{\sigma_t}{\alpha_t} \nabla_\theta D_{\text{KL}}\left(p_t(\boldsymbol{x}_t|\boldsymbol{x}_0 = \boldsymbol{g}_\theta(\boldsymbol{c}))||p_t(\boldsymbol{x}_t|y)\right)\right] \tag{6}$$

$$= \mathbb{E}_{\boldsymbol{\epsilon},\boldsymbol{c},t}\left[w_t\left(\boldsymbol{\epsilon}_\phi(x_t|y,t) - \boldsymbol{\epsilon}\right)\frac{\partial \boldsymbol{g}_\theta(\boldsymbol{c})}{\partial \theta}\right], \tag{7}$$

where $\epsilon_\phi$ is a text-to-image Diffusion Model and $y$ is the generation condition, e.g. text prompts. SDS needs to go through an optimization process on the 3D representation parameter $\theta$ to generate a single 3D model to generate 3D content.

Even though SDS can produce high-fidelity objects, there has been ongoing debate about the underlying theory. Recent works (Shi et al., 2023; Liang et al., 2023) also show that $L_{\text{SDS}}^\theta$ is equivalent to a reconstruction loss:

$$L_{\text{SDS}}^\theta = \mathbb{E}_{\boldsymbol{\epsilon},\boldsymbol{c},t}\left[\frac{1}{2}w_t \frac{\alpha_t}{\sigma_t}||\hat{\boldsymbol{x}}_t^{\text{c}} - \hat{\boldsymbol{x}}_t^{\text{gt}}||_2^2\right], \tag{8}$$

where $\hat{\boldsymbol{x}}_t^{\text{c}} = \boldsymbol{g}_\theta(\boldsymbol{c})$ and $\hat{\boldsymbol{x}}_t^{\text{gt}} = \frac{\boldsymbol{x}_t - \sigma_t \boldsymbol{\epsilon}_\phi(\boldsymbol{x}_t|y,t)}{\alpha_t}$ is the one-step "estimated ground-truth image" from Diffusion Models (i.e. sample-prediction), whose gradient is detached. Some other works (Yu et al., 2023; Katzir et al., 2023; Tang et al., 2023) also tried to explain SDS by analyzing the function of each component of SDS loss. In this paper, we will provide another interpretation of SDS: it can be viewed as a generalized DDIM generation process.

Another simple yet effective technique is to apply timestep annealing trick (Huang et al., 2023; Wang et al., 2024; Zhu & Zhuang, 2023), which can improve generation quality significantly. This technique is intuitive because reducing added noise during the latter stages of optimization, enabling models to discern finer details and iteratively improve upon them. Let us denote the time in the SDS optimization process as $\tau$ and the term that was taken expectation in the definition of SDS (Eq. 8) as $L_{\text{sds}}^\theta(\boldsymbol{\epsilon},\boldsymbol{c},t) = \frac{1}{2}\frac{\alpha_t}{\sigma_t}||\hat{\boldsymbol{x}}_t^{\text{c}} - \hat{\boldsymbol{x}}_t^{\text{gt}}||_2^2$. We use lowercase letters footnote for $L_{\text{sds}}^\theta$ to distinguish it from Eq. 8. Finally, the SDS optimization process with timestep annealing can be written as:

$$\frac{\mathrm{d}\theta}{\mathrm{d}\tau} = w_t \mathbb{E}_{\boldsymbol{\epsilon},\boldsymbol{c}}\left[\nabla_\theta L_{\text{sds}}^\theta(\boldsymbol{\epsilon},\boldsymbol{c},t = t(\tau))\right], \tag{9}$$

where $t(\tau)$ is a monotonically decreasing function of $\tau$.

## 3 FLOW SCORE DISTILLATION FOR 2D GENERATION

In this section, we only consider generation on 2D using SDS as SDS loss can also be applied to image representations. In this case, $\theta = \boldsymbol{g}_\theta(\boldsymbol{c})$ and $\frac{\partial \boldsymbol{g}_\theta(\boldsymbol{c})}{\partial \theta} = \boldsymbol{I}$. Then $L_{\text{sds}}^\theta$ becomes the following form:

$$\nabla_\theta L_{\text{sds-2d}}^\theta(\boldsymbol{\epsilon},t) = \boldsymbol{\epsilon}_\phi(x_t|y,t) - \boldsymbol{\epsilon}. \tag{10}$$

We will reveal a simple but profound connection between SDS and DDIM and give our interpretation of SDS in this section.

### 3.1 SIMPLIFIED FORMULATION OF DIFFUSION PF-ODE

We first reveal that PF-ODE (Eq. 5) can be formulated by an analogue of SDS (Eq. 10) in this section. We first define

$$\boldsymbol{x}_t = \alpha_t \hat{\boldsymbol{x}}_t^{\text{c}} + \sigma_t \tilde{\boldsymbol{\epsilon}}, \tag{11}$$

where $\tilde{\boldsymbol{\epsilon}}$ is a constant for each ODE trajectory. Notice that when $t = T$, the initial condition of the ODE gives $\tilde{\boldsymbol{\epsilon}} = 0 \cdot \hat{\boldsymbol{x}}_t^{\text{c}} + 1 \cdot \tilde{\boldsymbol{\epsilon}} = \boldsymbol{x}_T \sim \mathcal{N}(\boldsymbol{0}, \boldsymbol{I})$. Intuitively, $\tilde{\boldsymbol{\epsilon}}$ **can be viewed as the noise added to** $\hat{\boldsymbol{x}}_t^{\text{c}}$**, and** $\hat{\boldsymbol{x}}_t^{\text{c}}$ **as the clean image at timestep** $t$. So we will also refer to $\tilde{\boldsymbol{\epsilon}}$ as *the initial noise* apart from *the added noise* in this paper. It is noteworthy that the concept of the clean image $\hat{\boldsymbol{x}}_t^{\text{c}} = \frac{\boldsymbol{x}_t - \sigma_t \tilde{\boldsymbol{\epsilon}}}{\alpha_t}$ is different from the aforementioned estimated ground-truth image $\hat{\boldsymbol{x}}_t^{\text{gt}} = \frac{\boldsymbol{x}_t - \sigma_t \boldsymbol{\epsilon}_\phi(\boldsymbol{x}_t|y,t)}{\alpha_t}$. By applying "*change-of-variable*" trick and change the variable of the Diffusion PF-ODE from $\boldsymbol{x}_t$ to $\hat{\boldsymbol{x}}_t^{\text{c}}$, we have:

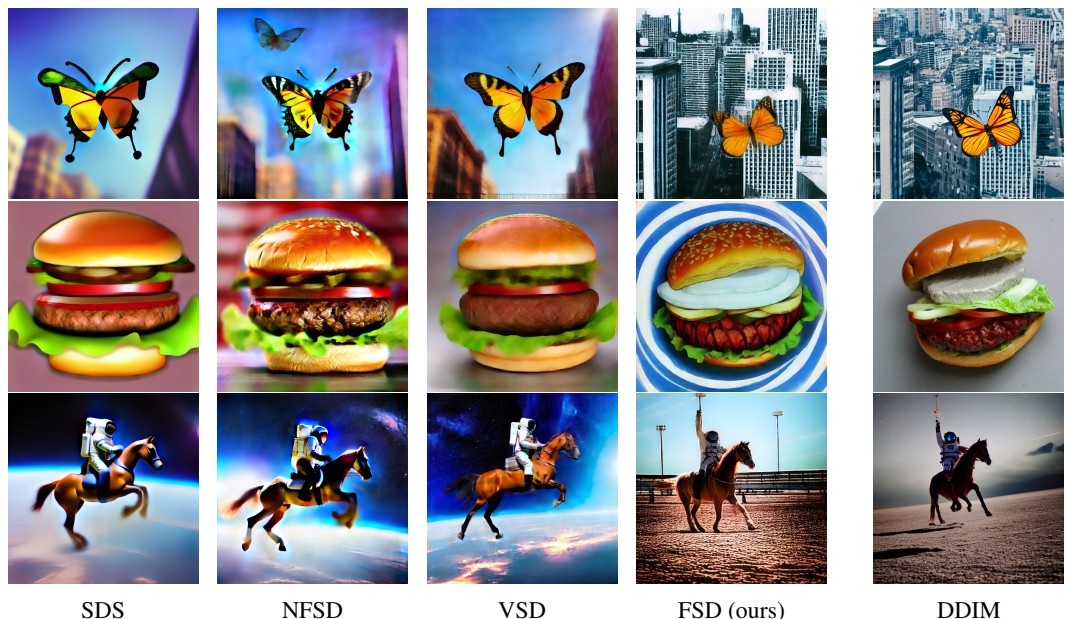

|  SDS | NFSD | VSD | FSD (ours) | DDIM |

Figure 3: **Generation results of different methods on image space with the same random seeds.** FSD can generate images that are very similar to images generated by DDIM given the same initial noise (implied by Prop. 1). However, FSD can also be used for 3D generation, a task for which DDIM is not suitable. See experiment details in Appx. Sec. E.2.

**Proposition 1** (An equivalent form of Diffusion PF-ODE). *Diffusion PF-ODE (Eq. 5) can be equivalently formulated by an analogue of SDS loss (Eq. 10):*

$$\frac{\mathrm{d}\hat{\boldsymbol{x}}_t^c}{\mathrm{d}t} = \frac{\mathrm{d}(\sigma_t/\alpha_t)}{\mathrm{d}t}\left[\boldsymbol{\epsilon}_\phi(\boldsymbol{x}_t|t,y) - \tilde{\boldsymbol{\epsilon}}\right] \tag{12}$$

$$= w_t' \nabla_\theta L_{sds\text{-}2d}^\theta(\tilde{\boldsymbol{\epsilon}}, t), \tag{13}$$

*where $\boldsymbol{x}_t = \alpha_t \hat{\boldsymbol{x}}_t^c + \sigma_t \tilde{\boldsymbol{\epsilon}}$, $\theta = \hat{\boldsymbol{x}}_t^c$ and $w_t' = \frac{\mathrm{d}(\sigma_t/\alpha_t)}{\mathrm{d}t}$ is a weighting scalar.*

Please refer to Appx. Sec. F.3 for detailed derivation of this proposition. We also visualize the "change-of-variable" in Appx. Sec. G. Remarkably, in the context of image generation, we observe that the evolution direction of PF-ODE aligns precisely with the gradient of the SDS loss (Eq. 10) with fixed noise.

## 3.2 FLOW SCORE DISTILLATION ON 2D

Even though we found the evolution direction of Diffusion PF-ODE (Eq. 12) is very samilar to the SDS loss, there exist some notable differences compared to the original definition of SDS loss (Eq. 6). Specifically, i) the timestep in a DDIM process is monotonically decreasing, aligning with the timestep annealing technique (Wang et al., 2024; Zhu & Zhuang, 2023; Huang et al., 2023). But SDS uses randomly sampled timestep. ii) And the change-of-variable trick (Eq. 11) we used during our simplification process implies we should also add the same noise $\tilde{\boldsymbol{\epsilon}}$ throughout the SDS generation process, to align it with DDIM. In contrast, the original SDS uses random noise.

As we will demonstrate in subsequent sections and through our experiments, the second difference between DDIM and SDS significantly influences generation diversity. Therefore, we term our approach that combines **timestep annealing and consistent noise sampling strategy throughout the generation process** as *Flow Score Distillation* (FSD) to differ it from SDS. We visualize image generation results using several SDS-like methods, FSD and DDIM in Fig. 3 to demonstrate the differences between SDS-based methods and FSD. We summary the difference between FSD, SDS, and DDIM when applied to 2D images in the following Tab. 1.

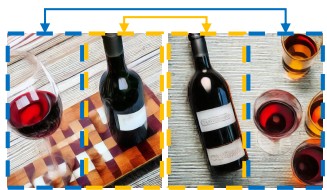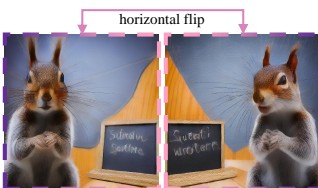

Figure 4: **Impact of initial noise $\tilde{\epsilon}$.** Experiments show that the local textures of noise added during FSD optimization are highly correlated with the textures of the final image. We shuffle the patches of initial noise $\tilde{\epsilon}$ used by FSD and observe that the textures of generated images are shuffled in the same way. This property inspired our design of world-map noise function $\epsilon(c)$ for 3D generation in this work. In this figure, the parts framed by dotted lines of the same color share the same initial noise $\tilde{\epsilon}$ patches.

### 3.3 Analysis of the Noise Sampling Strategy

Our empirical investigation reveals that the generation results produced by SDS exhibit an undesirable tendency toward over-smoothness and lack of diversity. Remarkably, even with the timestep annealing technique, this issue persists. This tendency originates from the maximum-likelihood-seeking nature implied by its definition (Eq. 6) where it models the current distribution using a Dirichlet function centered at $g_\theta(c)$. As a result, the generation results of SDS will be centered around a few modes (Poole et al., 2022), i.e. a few maximum likelihood points on the smoothed distribution $p_t$. In contrast, not only does FSD yield diverse outcomes, but it can also generate highly detailed samples. This can be attributed to that FSD is more aligned with a DDIM process, which can generate samples from exactly the target distribution $p_0$. So we conclude that the noise sampling strategy can affect the generation diversity greatly.

## 4 Lifting Flow Score Distillation to 3D

As highlighted in Sec. 3.3, we have identified that the noise sampling strategy might contribute to the decline of the diversity of SDS significantly. Building upon this insight, we follow the discussion of FSD in Sec. 3.2 and propose to use deterministic noise generation strategy. We can directly generalize FSD to arbitrary 3D representations $g_\theta(c)$:

$$\nabla_\theta L_{\text{FSD}}^\theta = \mathbb{E}_c \left[ \nabla_\theta L_{\text{sds}}^\theta(\epsilon = \epsilon(c), c, t = t(\tau)) \right] \tag{14}$$

$$= \mathbb{E}_c \left[ (\epsilon_\phi(x_t|y, t(\tau)) - \epsilon(c)) \frac{\partial g_\theta(c)}{\partial \theta} \right], \tag{15}$$

where $x_t = \alpha_t g_\theta(c) + \sigma_t \epsilon(c)$, $\epsilon(c)$ is a deterministic noise function generated at the beginning of the optimization and $t(\tau)$ is a monotonically decreasing timestep schedule function to optimization time $\tau$. Compared with original SDS loss, we do not take expectation on timestep $t$ and noise $\epsilon$ since $t$ is determined by $t(\tau)$ and noise function is deterministic. With this deterministicity requirement, fixed noise is always added to the same camera view, aligning with a DDIM process. We do not specify the form of the deterministic noise function $\epsilon(c)$ in FSD. However, we propose some rules for designing $\epsilon(c)$ based on the actual generation effect in Appx. Sec. I, according to our practical experiences. We will also introduce the *world-map noise function* as $\epsilon(c)$ for our experiments of 3D generation with FSD in this paper.

| | Noise Sampling | Optimizer | Timestep Schedule |
|---|---|---|---|
| SDS | random | Adam | random |
| FSD (ours) | fixed | Adam | annealing |
| DDIM | fixed | first-order discretization | annealing |

Table 1: Comparison of different methods for 2D image generation.

Alternatively, FSD loss can be seen as applying DDIM generation process on 3D representations through Jacobian of a differentiable renderer. This viewpoint shares some similarities to the interpretation of SDS from SJC (Wang et al., 2023a), who consider SDS as back-propagating the score (Song et al., 2020b) of Diffusion Models through Jocabian of the renderer. However, our interpretation offers more precise explanation on the relation between SDS and DDIM process (Prop. 1), in contrast to approximated 3D score function interpretation in SJC. Meanwhile, $\epsilon(c)$ is a deterministic noise function that generates correlated noise between views, which aligns with the prior that the nearby views are correlated. The design of FSD ensures the optimization directions are consistent, particularly when similar $c$s are sampled. As the $\epsilon(c)$s are similar, the generated ground-truth images should be consistent as well. Notably, recent works (Ge et al., 2023; Qiu et al., 2023; Chang et al., 2024) on video generation also find using designed video noise prior can improve the capabilities of Video Diffusion Models.

### 4.1 Designing $\tilde{\epsilon}$.

#### 4.1.1 Failure of a Vanilla Design of $\tilde{\epsilon}$

A vanilla design of $\epsilon(c)$ can be $\epsilon(c) = \epsilon$, which is a constant function. However, according to our experiments on text-to-3D generation, such a design can lead to poor geometry of the generated samples. Typically, holes on the surfaces are observed (see Appx. Sec. B.3). We attribute this effect to the uneven convergence speed of FSD in 3D space caused by the constant noise function. Our experiments on 2D show that the local textures of noise added during FSD optimization are highly correlated with the local textures of the final image (Demonstrated in Fig. 4). And in text-to-3D generation with FSD, the generated ground-truth images have more consistent textures at the center point than other points in 3D space, due to the sampling strategy of camera view $c$. As a result, the convergence speed at the center point is much higher than at other points, leading to holes on the surfaces. Flaws are also observed in Video Diffusion Models that adopt fixed noise prior, due to similar reasons, which is known as the textures sticking problem (Chang et al., 2024).

#### 4.1.2 World-map Noise Function $\tilde{\epsilon}$.

Even through directly apply constant noise could result in degraded geometry, it's no-trivial to design a view dependent noise function due to the special property of Gaussian noise. To avoid relating specific noise textures to specific points in 3D space throughout the generation process but still augment the consistency of added noise between camera views, we propose *world-map noise function* $\epsilon(c)$ in this paper (methods visualized in Fig. 2), which aligns noise textures coarsely in 3D space while avoiding converging too fast at a specific point in 3D space. Furthermore, we show that our methods can be seen as aligning noise on a sphere in Appx. Sec. I.

Specifically, we first compute a foreground mask $M(r) = \mathbf{1}_{\alpha(\mathrm{r}) > \alpha_0}$ for each ray $r$, where $0 \leq \alpha(r)$, $\alpha_0 \leq 1$ is the opacity of pixel $r$. We add the same noise for the background and query a noise patch from the noise world-map $\epsilon_p$ of size $D \times H \times W$ for the foreground. Let us denote the camera parameters sampled on the sphere as $c = (\mathrm{FOV}, r_{\mathrm{cam}}, \theta_{\mathrm{cam}}, \phi_{\mathrm{cam}})$. Both $\epsilon_b$ and $\epsilon_p$ are sampled from $\mathcal{N}(\mathbf{0}, \mathbf{I})$ at the beginning of the optimization. The world-map noise function $\epsilon(c)$ is:

$$\epsilon(c) = (1 - M) \odot \epsilon_b + M \odot W_{(W \frac{\phi_{\mathrm{cam}}}{2\pi}, H \frac{\theta_{\mathrm{cam}}}{\pi})}(\epsilon_p), \tag{16}$$

where $W_{(W \frac{\phi_{\mathrm{cam}}}{2\pi}, H \frac{\theta_{\mathrm{cam}}}{\pi})}(\epsilon_p)$ operation refers to the noise patch of size $D \times H_{\mathrm{hidden}} \times W_{\mathrm{hidden}}$ centered at position $(W \frac{\theta_{\mathrm{cam}}}{2\pi}, H \frac{\phi_{\mathrm{cam}}}{\pi})$ on the noise worldmap $\epsilon_p$. We visualize this noise map in the accompanying video in project page. We also provide a pseudocode of our algorithm in Appx. Sec. H.

### 4.2 Coarse to Fine Pipeline

SDS like methods usually suffers from multi-face problems. Even though our methods provides consistent guidance for the same camera view, FSD may still suffer from multi-face problems since the noise does not provide camera position related information. We tackle this issue by using text-to-multiview-image diffusion model (Shi et al., 2023) that is trained on 3D dataset (Deitke et al., 2023). Specifically, we distill MVDream with FSD in the first stage to generate a coarse shape. Even though the generated shapes are usually free of multi-face problems, the colors of the objects are usually unnatural. This is mainly because MVDream usually generates multi-view images with unnatural

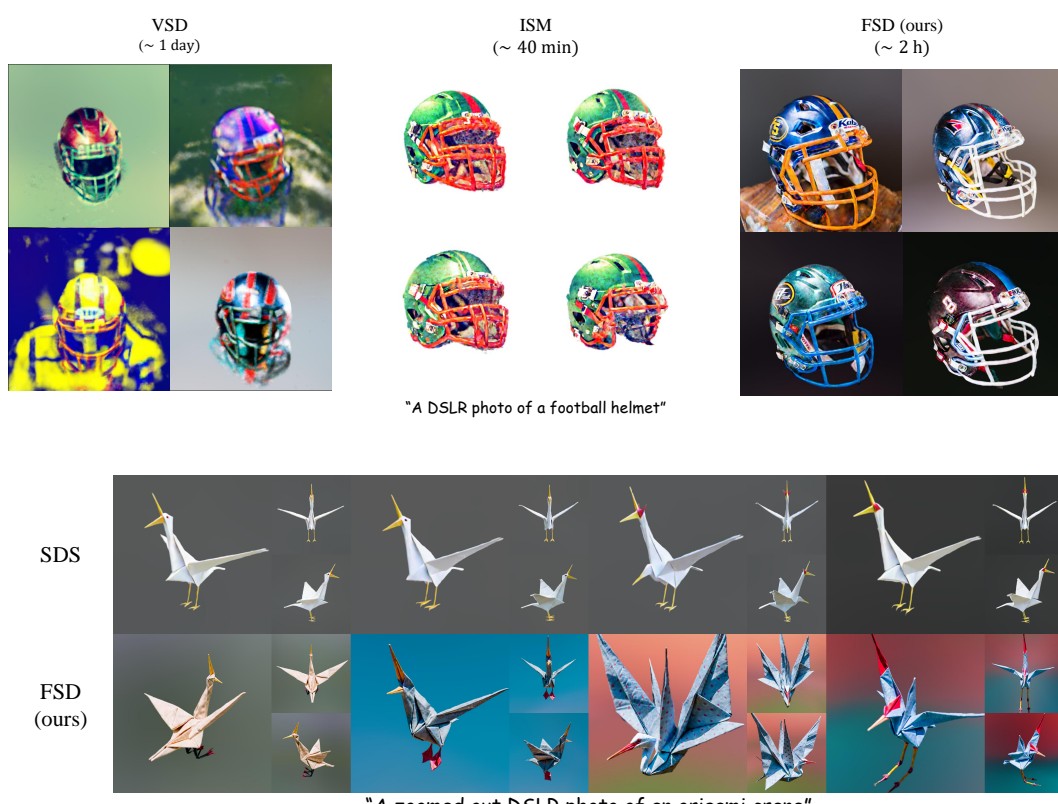

Figure 5: **Comparisons to baseline on text-to-3D Generaion.** Our method can generate diverse 3D models with realistic and detailed appearances. We compare our method with the baseline including VSD (ProlificDreamer) (Wang et al., 2024) and ISM (LucidDreamer) (Liang et al., 2023). We set particle number to 4 for VSD. We use 4 different random seeds for other methods. Our method can generate high quality and detailed objects in reasonable time.

colors even with DDIM. We propose to further refine the generated shape with FSD distilling Stable Diffusion in the second stage to refine the color and details.

### 4.3 COMPARE FSD WITH SDS ON 3D

Apart from timestep annealing trick (Huang et al., 2023; Zhu & Zhuang, 2023; Wang et al., 2024), FSD is different from SDS in terms of noise sampling strategy as well. In case when the same camera view $c$s are sampled, FSD yields consistent one-step estimated ground-truth images since $\epsilon(c)$ is the same. Even when different camera views $c$s are sampled, the ground-truth images are still coarsely aligned. In contrast, one can see SDS as using an uncorrelated noise prior $\epsilon(c)$ on $c$, which always yields ground-truth images that are inconsistent and have notable differences. We also discuss the relation between our method with recent works (Wu et al., 2024; Gu et al., 2023) in Appx. Sec. D.

## 5 EXPERIMENTS

### 5.1 IMPLEMENTATION DETAILS

To control for variables, our quantitative experiments are conducted with the threestudio codebase (Guo et al., 2023). We apply timestep annealing for all baseline methods. We use official implementations of baseline methods in qualitative comparison. We use random seeds from 0 to 3 for each prompt by default to demonstrate the diversity of generated samples. Please refer to Appx. Sec. E.1 for more implementation details.

## 5.2 Evaluation on Text-to-3D Generation

### 5.2.1 Qualitative Comparison.

We visualize the experiment results of SDS (Shi et al., 2023; Wang et al., 2023a), VSD (Wang et al., 2024), ISM (Liang et al., 2023) and our FSD in Fig 5. VSD incorporated LoRA finetuning in the generation process and replace the random noise term in SDS with prediction from the LoRA network. ISM Incorporated DDIM inversion noising into their optimization to enhance guidance consistency. Compared with baseline methods, our FSD can generate diverse and detailed objects in reasonable time. We also provide additional qualitative comparison in Appx. Sec. A.1.

### 5.2.2 Quantitative Results.

**3D-FID** We compute the FID score between the rendered images of the 3D objects and the images generated by DDIM following VSD (Wang et al., 2024). We collected 5,000 images per prompt from Stable Diffusion for each of the 10 randomly selected prompts, forming a real image set of 50,000 images in total. Using various score distillation methods, we generated 3D models with 10 distinct seeds for each method. Each 3D object was rendered from 60 different angles to reduce the variance of FID metric, producing a fake image set of 6,000 images. The results are shown in Tab. 2. We provide additional measurement on diversity in Appx. Sec. C.

|  | 3D-FID $\downarrow$ |
|---|---|
| SDS (Poole et al., 2022) | 88.06 |
| ISM (Liang et al., 2023) | 86.00 |
| VSD (Wang et al., 2024) | 83.02 |
| FSD (ours) | **78.75** |

Table 2: We compare the generation quality and diversity in this experiment.

### 5.3 Ablation Study

We provide additional ablation study on our proposed coarse-to-fine pipeline, noise function and hyper parameters for noise function in Appx. Sec. B.

## 6 Conclusion

In this work, we systematically study the problem of text-to-3D generation. We first review the theorems of SDS and reveal a simple but profound underlying connection between DDIM and SDS. Following this insight, we propose FSD to tackle the diversity degradation challenge. By using a consistent noise sampling schedule that aligns noise coarsely in 3D space, FSD breaks free from the maximum-likelihood-seeking nature of SDS and could generate diverse results with high quality. Additional, our methods incorporate a multiview diffusion model that has rich shape prior in coarse stage and a image diffusion model that has rich texture prior in refine stage, our methods can generate diverse and high quality 3D objects.

**Limitations and future works.** Although FSD could improve the diversity and quality of 3D generation, we found that it is still difficult to generate 3D models as diverse as the images generated through DDIM generation process. We believe this mainly originates from our direct generalization from 2D DDIM to 3D. The deterministicity requirement on noise function only make sure the update is aligned with DDIM process when only a single camera view is considered. It can be hard to guarantee that the DDIM trajectory is followed exactly, especially when the updates from other camera views are considered. Second, even through our proposed noise function provided more aligned guidance across camera views, hindered by the special property of Gaussian noise, we only find a design of worldmap noise map function that only aligns noise on a sphere independent of object surface. This misalignment could potently hinder the performance of FSD and may not work for objects with complex geometry. We do not specify the noise function $\epsilon(c)$ in the general form of FSD (Eq. 15), and better designs of $\epsilon(c)$ may exist. Third, like other score distillation methods, our method can still suffer from multi-face problems. Seeking help from multi-view image diffusion models that are trained on limited amount of 3D data, our 2 stage pipeline can generate shapes with high success rate. But our 2 stage pipeline may not work for complex prompts due to the limited ability of the multi-view diffusion model. Lastly, like other score distillation methods, the generation of our methods may take several hours.

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

# APPENDIX

## A ADDITIONAL RESULTS

### A.1 ADDITIONAL COMPARISON WITH BASELINES

VSD                                    FSD (ours)

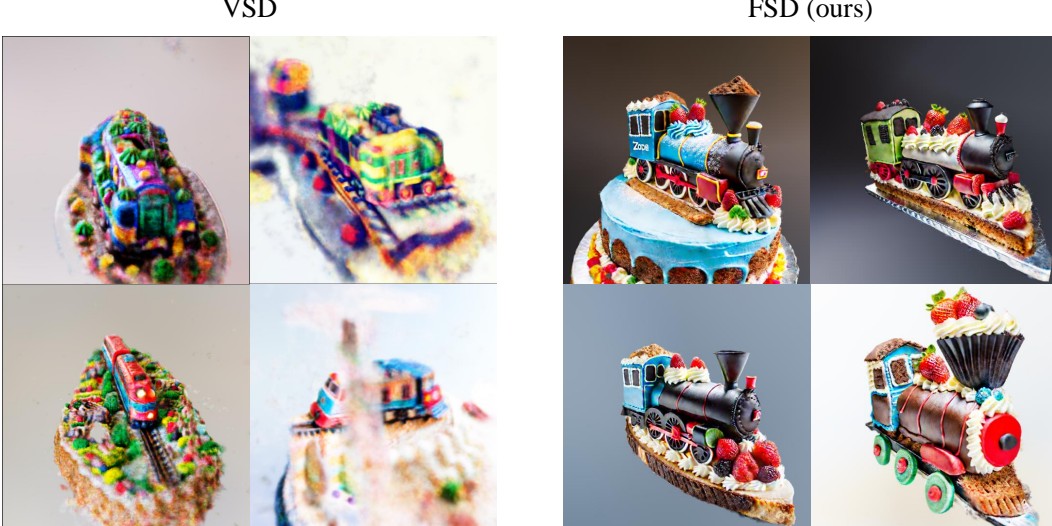

"A zoomed out DSLR photo of a cake in the shape of a train"

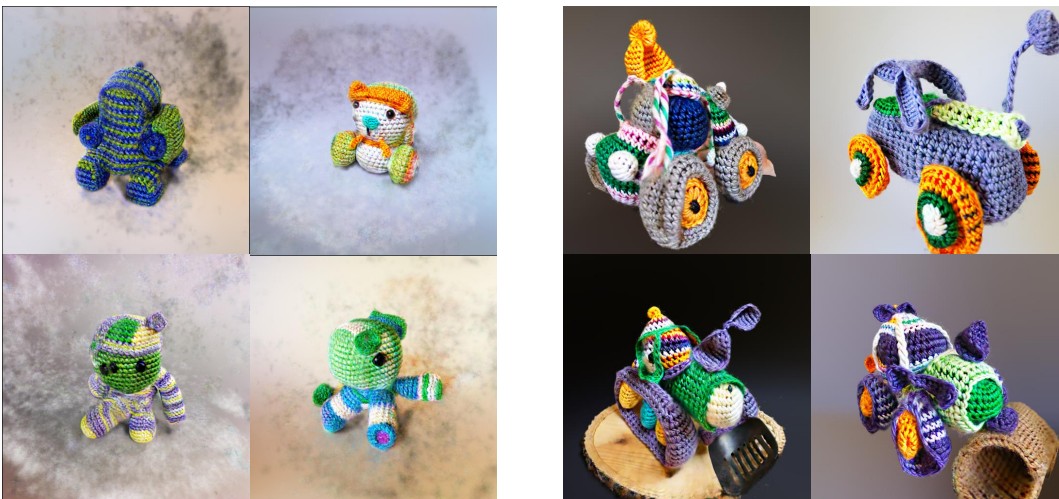

"An amigurumi bulldozer"

Figure 6: **Comparison with VSD.** We compare with VSD (Wang et al., 2024). We set particle number to 4 for VSD, we use 4 random seeds for our FSD.

We show qualitative results in this section, we compare our FSD with VSD (Wang et al., 2024), ISM (Liang et al., 2023) and SDS (Poole et al., 2022) in this experiment. We use official code implementation of VSD (Wang et al., 2024), ISM (Liang et al., 2023) in this comparison (Fig. 6 and Fig. 7). For SDS (Poole et al., 2022), we use the code implementation of MVDream (Shi et al., 2023) (Fig. 8 and Fig. 9). Our methods can generation diverse and high quality results in reasonable time (∼2 h on A100) compared with 4 particle VSD (∼ 1 day on A100). Our methods can generate 3D objects with higher quality and diversity compared with ISM and SDS.

ISM                                                      FSD (ours)

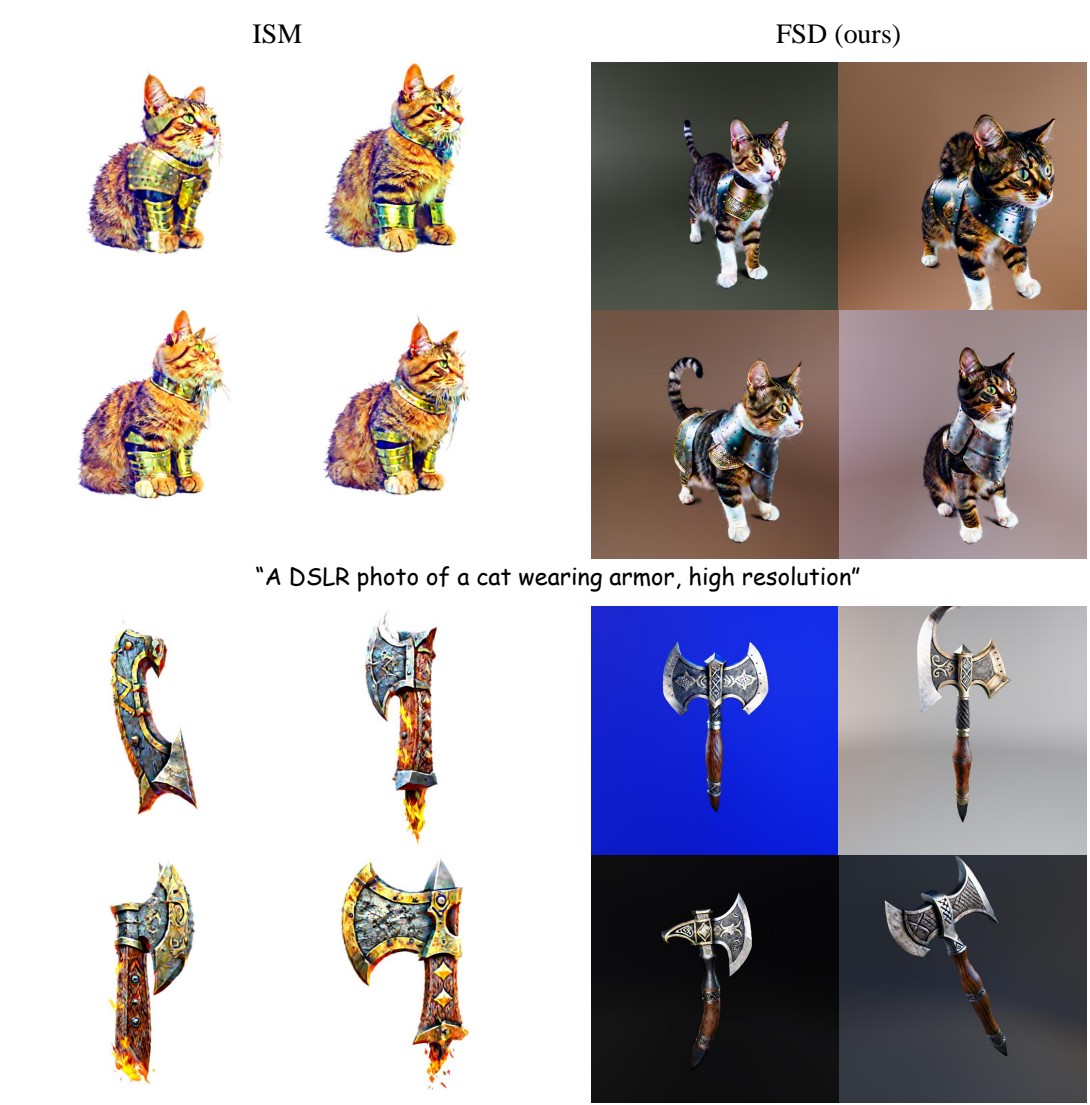

"A DSLR photo of a cat wearing armor, high resolution"

"Viking axe, fantasy, weapon, blender, 8k, HDR"

Figure 7: **Comparison with VSD.** We compare with ISM (Liang et al., 2023). We use 4 random seeds for comparison.

## A.2    IMAGE TO 3D GENERATION

We use zero123-xl (Liu et al., 2023a) in this experiment. By applying our worldmap noising, the backview of object are more diverse and can form finer details. We visualize the frontview and generated results with SDS and our FSD in Fig. 10.

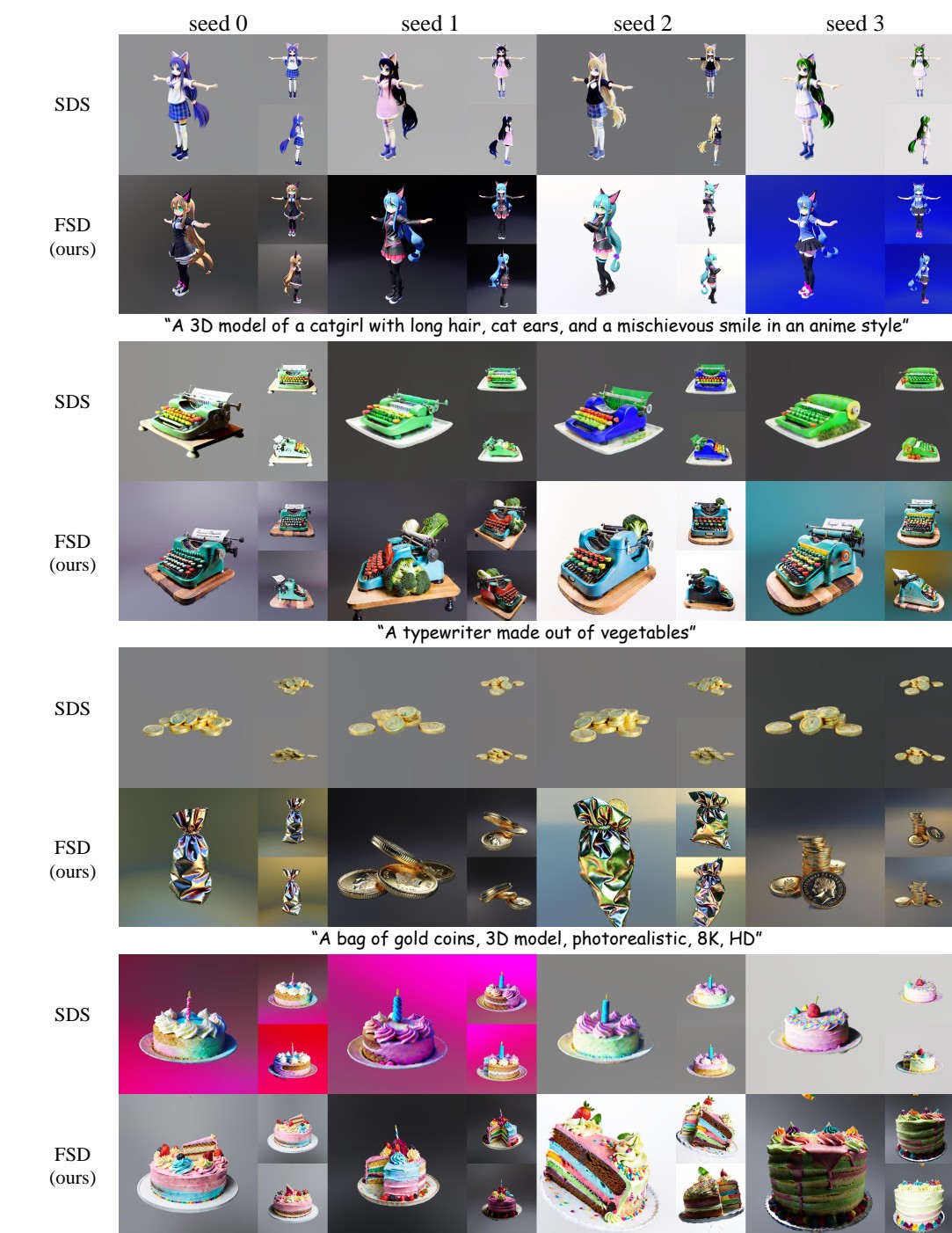

Figure 8: **Comparison with SDS.** We compare with SDS (Poole et al., 2022). We use 4 random seeds for comparison.

## A.3 ADDITIONAL 2D EXPERIMENTS ON NOISE PRIOR

We provided additional generation results of FSD on 2D image space with shuffled or flipped initial noise in 11. The patches framed with the same color in the same row share the same initial noise patches $\tilde{\epsilon}$.

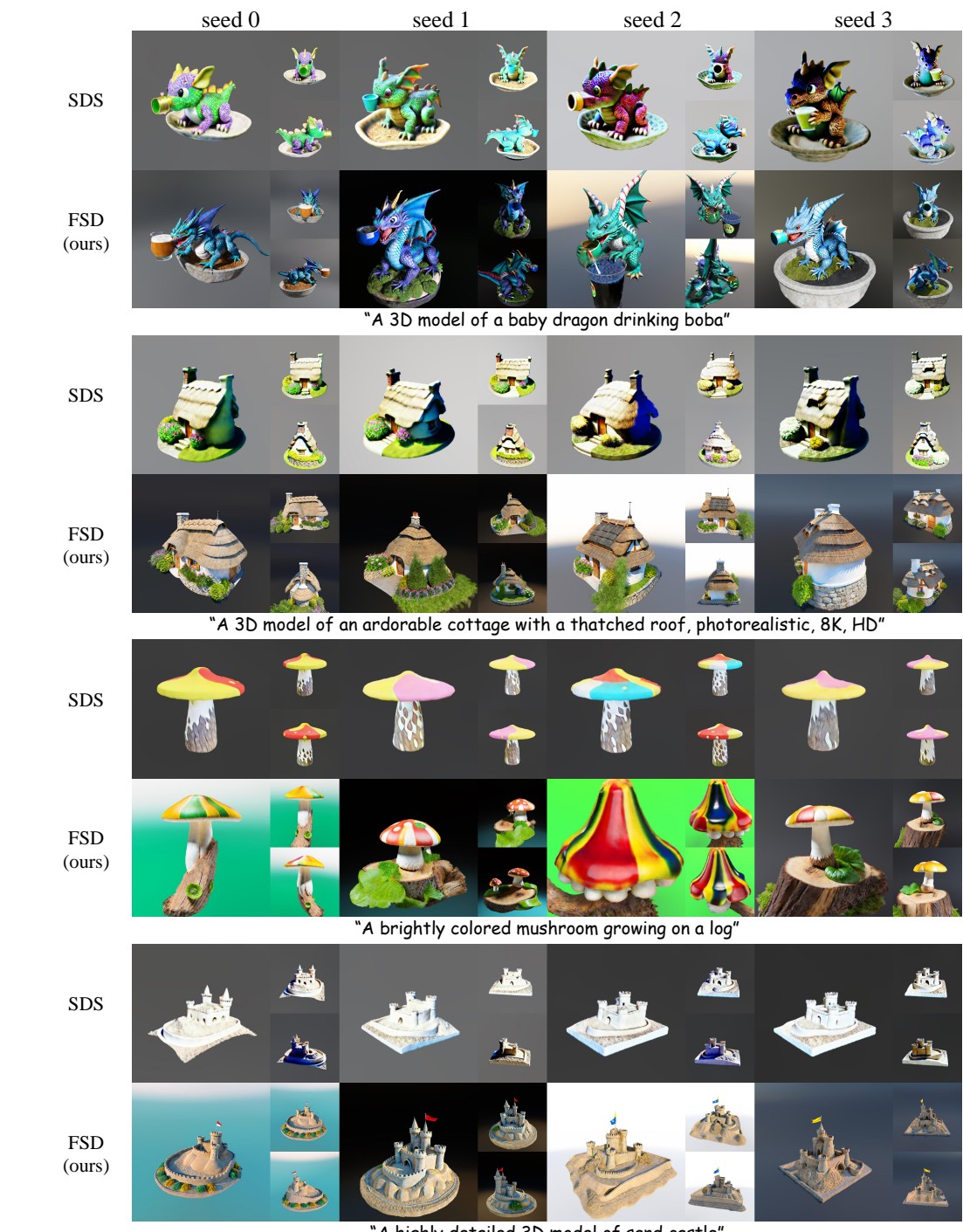

Figure 9: **Comparison with SDS.** We compare with SDS (Poole et al., 2022). We use 4 random seeds for comparison.

972
973
974
975
976
977
978
979
980
981
982
983
984
985
986
987
988

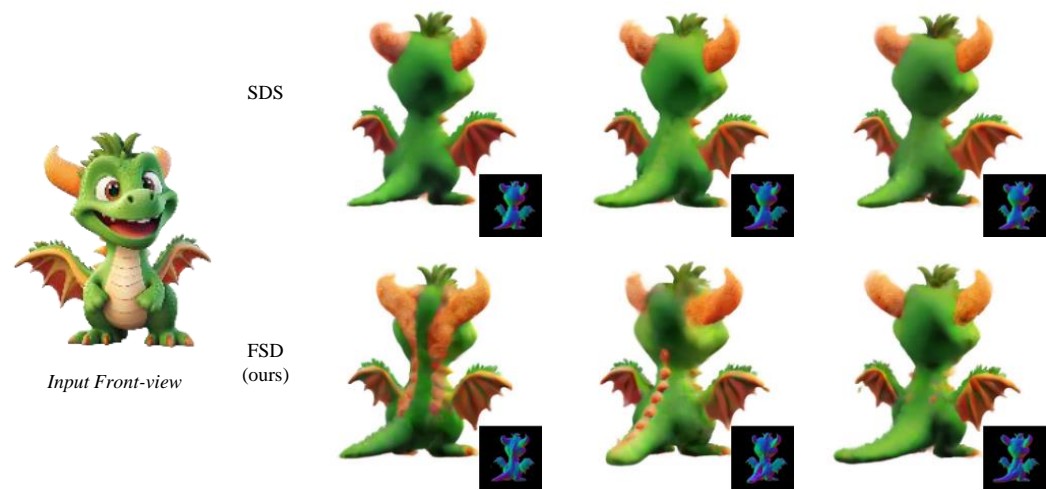

Figure 10: **Image to 3D Generation.** We use Zero123-xl in this experiment. We compare our methods FSD with SDS. Our method can generate diverse backview of the object.

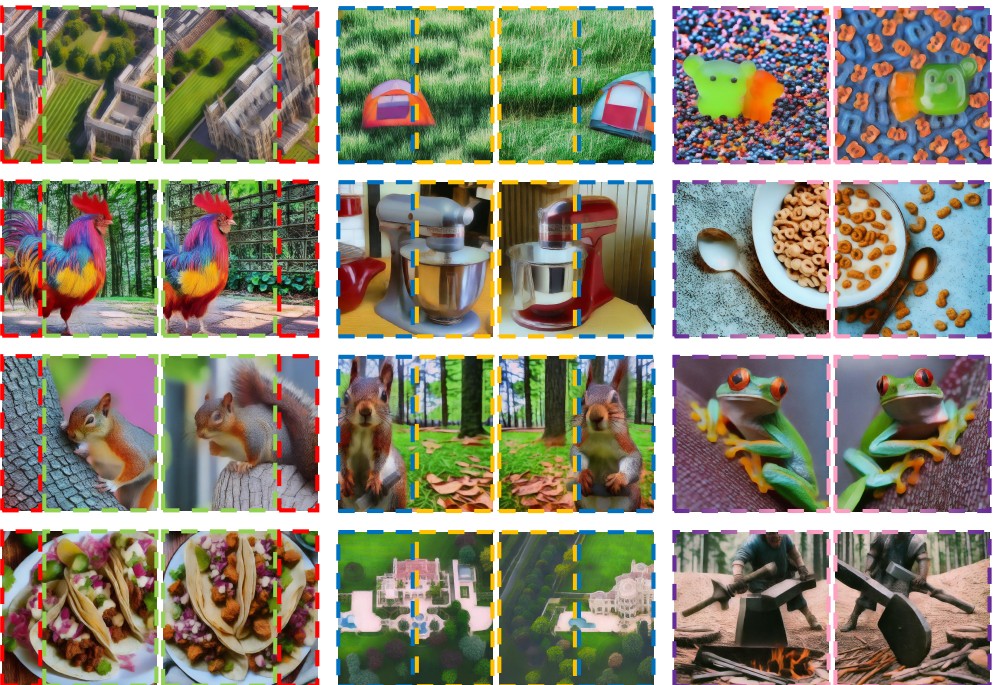

Figure 11: **Impact of initial noise $\tilde{\epsilon}$.** We provide additional 2D experiment results generated by FSD show the impact of the initial noise $\tilde{\epsilon}$. The patches framed with the same color in the same row share the same initial noise patches.

## B  ADDITIONAL ABLATIONS

### B.1  ABLATION ON PROPOSED PIPELINE

We propose to use a 2 stage pipeline to generate 3D objects with FSD. We use text-to-multiview-image diffusion model MVDream in the first stage to generate coarse shape to avoid multi-face problems. We use text-to-image diffusion model Stable Diffusion to refine the generated colors and details in the second stage. We visualize the results in Fig. 12.

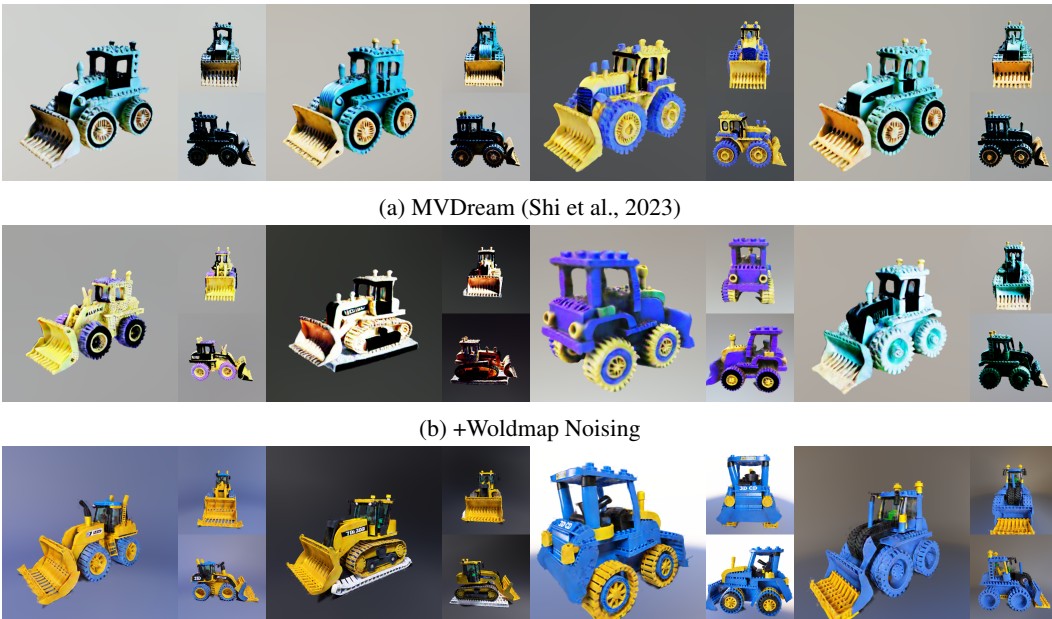

(a) MVDream (Shi et al., 2023)

(b) +Woldmap Noising

(c) +Stable Diffusion Refinement (stage 2)

Figure 12: **Ablation on our proposed pipline.** (a) MVDream applies SDS (Poole et al., 2022) to generate 3D shape, but results are similar with different random seeds. (b) Adding Worldmap noise make the results more diverse, but due to limited abality of teacher diffusion model, the 3D objects are lack of details and colors are unnatural. (c) With additional refinement stage with Stable Diffusion, the objects form more details. The prompt for this figure is "A 3D model of a bulldozer made out of toy bricks".

## B.2 NOISE INTERPOLATION

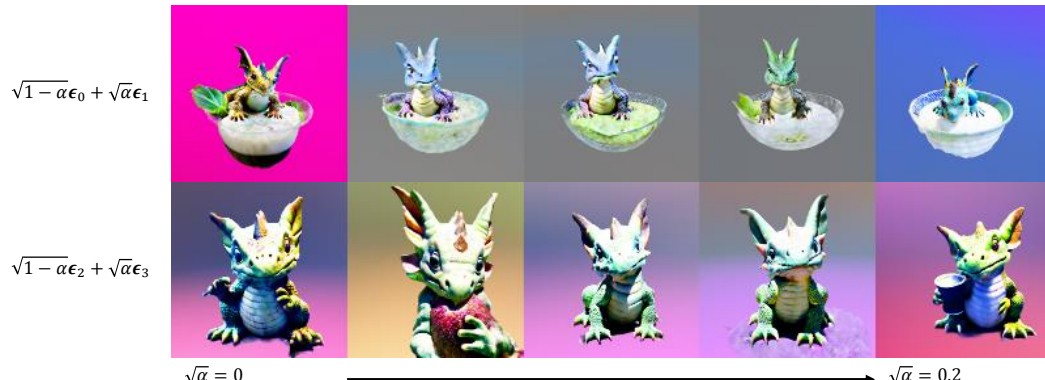

$\sqrt{1-\alpha}\epsilon_0 + \sqrt{\alpha}\epsilon_1$

$\sqrt{1-\alpha}\epsilon_2 + \sqrt{\alpha}\epsilon_3$

$\sqrt{\alpha} = 0$ $\qquad\longrightarrow\qquad$ $\sqrt{\alpha} = 0.2$

Figure 13: **Noise interpolation.** We show that initial noise can control generation results in this figure. The two rows correspond to 2 different noise main components $\epsilon_0$ and $\epsilon_2$. We show the results of the coarse stage in our pipeline. The prompt for this figure is "A baby dragon drinking boba".

Since the initial noise can be viewed as the identity of the generated object, we show that initial noise can control the generation results in this experiment. Starting from two different initial noise $\epsilon_0$ and $\epsilon_2$, the generated results are very different. While gradually blending small amount of new noise component into the initial noises, the generated results remain mostly unchanged.

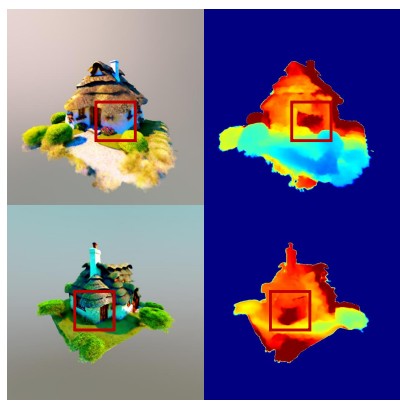 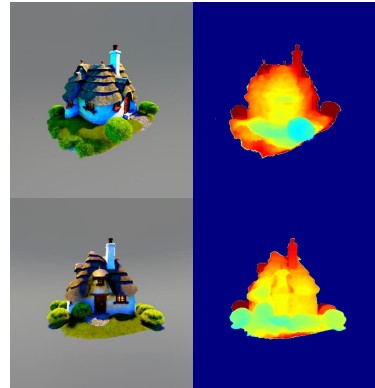

Vanilla Constant Noise Function $\epsilon(c)$                    World-map Noise Function $\epsilon(c)$ (proposed)

Figure 14: **Compare world-map noise function $\epsilon(c)$ with a vanilla design of constant function $\epsilon$.** We visualize the rendered images and depth maps for the two noise methods. The vanilla design of noise function $\epsilon$ can easily lead to holes on the surfaces (framed in red), even when the RGB images seem plausible. In contrast, no obvious flaws are observed in the results of FSD with the world-map noise function.

### B.3 ABLATION ON NOISE FUNCTION

We also provide an additional comparison between the vanilla constant noise function and the world-map noise function in Fig. 14. We find the constant noise function may harm the geometry of the generation results.

### B.4 ABLATIONS ON HYPER PARAMETERS

We use a parameter $\Theta$ to control the noise world-map's size ($H$ and $W$). $H$ and $W$ are determined by $\Theta$ according to $H = H_{\text{hidden}} \frac{\pi}{\Theta}$ and $W = W_{\text{hidden}} \frac{2\pi}{\Theta}$. We visualize the results corresponding to different $\Theta$s in Fig. 15.

## C ADDITIONAL QUANTITATIVE RESULTS

We compute several metrics for 3D generation results with SDS (Poole et al., 2022; Wang et al., 2023a) and FSD. When using stable diffusion (Rombach et al., 2022) as backbone, we use 16 prompts and 4 random seeds for each prompt (Tab. 3). When using MVDream (Shi et al., 2023) as backbone, we also use 16 prompts and 4 random seeds for each prompt (Tab. 4).

Table 3: stable diffusion (Rombach et al., 2022) as backbone

| Method | 3D-CLIP ($\uparrow$) | 3D-IS ($\uparrow$) | CROSS-FID ($\uparrow$) |
|---|---|---|---|
| DDIM Images | $33.72 \pm 1.83$ | $1.68 \pm 0.55$ | - |
| SDS | $32.57 \pm 1.43$ | $1.58 \pm 0.47$ | $106.5 \pm 58.3$ |
| FSD (ours) | $\mathbf{32.72 \pm 1.56}$ | $\mathbf{1.78 \pm 0.49}$ | $\mathbf{141.8 \pm 57.9}$ |

Table 4: MVDream (Shi et al., 2023) as backbone

| Method | 3D-CLIP ($\uparrow$) | 3D-IS ($\uparrow$) | CROSS-FID ($\uparrow$) |
|---|---|---|---|
| DDIM Images | $34.64 \pm 2.56$ | $2.02 \pm 0.47$ | - |
| SDS | $\mathbf{30.93 \pm 3.40}$ | $1.77 \pm 0.37$ | $86.6 \pm 33.4$ |
| FSD (ours) | $30.12 \pm 3.07$ | $\mathbf{2.13 \pm 0.35}$ | $\mathbf{174.8 \pm 44.5}$ |

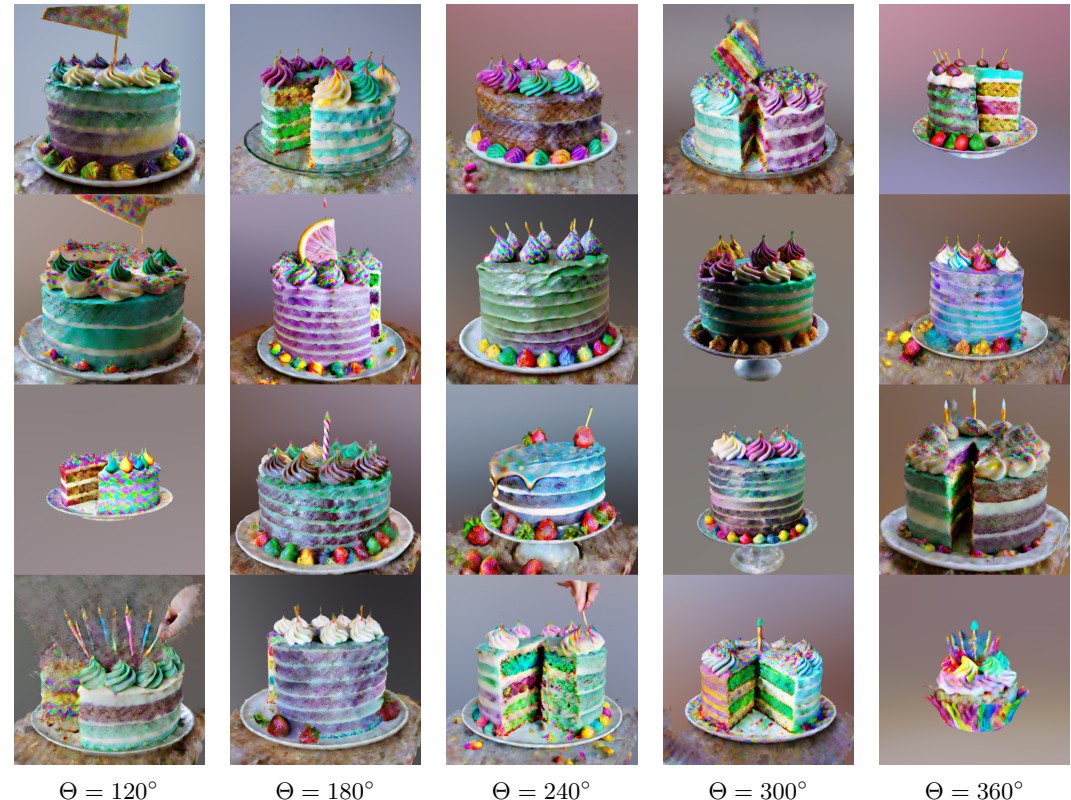

$$\Theta = 120° \qquad \Theta = 180° \qquad \Theta = 240° \qquad \Theta = 300° \qquad \Theta = 360°$$

Figure 15: **Ablation on other hyperparameters in $\epsilon(c)$.** We use a parameter $\Theta$ to control the size of noise world map ($H$ and $W$) in $\epsilon(c)$. When $\Theta$ is larger, the "radius $r_+$" (Eq. 29 and Eq. 30) of the noise world map is smaller and FSD trends to generate smaller 3D models. In practice, we found FSD is prone to the parameter $\Theta$.

**3D-CLIP** We compute CLIP score (Hessel et al., 2021; Radford et al., 2021) using ViT-B/32 to measure the semantic similarity between the renderings of the generated 3D object and the input text prompt. We sample 24 views for each prompt and each seed when computing CLIP score.

**3D-IS** We compute IS score (Salimans et al., 2016) to measure both the image quality and diversity. We first compute the IS scores of sampled views for each prompt and then average the IS scores across prompts.

**CROSS-FID score** To directly measure the diversity of generation results, it is natural to measure the inception distance between different generated samples. We first sample 24 views for each prompt and each seed. Then we separate the images corresponding to random seeds 0, 1 and 2, 3 into two sets of images. We compute FID (Heusel et al., 2017) of the two sets of images and average the FID score across prompts. We term this score as CROSS-FID score since it is different from the standard way of using FID to evaluate GANs (Heusel et al., 2017) and 3D-FID in Sec. 5.2.2.

## D    Discussions

**Difference with Consistent3D** Recent work Consistent3D (Wu et al., 2024) also applied fixed noise when conducting SDS-like generation. In Consistent3D, they follow the idea of Consistent Training (Song et al., 2023) and use the rendered image perturbed with fixed noise to approximate the starting point of the deterministic flow. In our method, for the same camera view, we also add fixed noise to the rendered image, but the noised image is used to simulate a variable in the middle of a PF-ODE trajectory, which is different from Consistent3D. Our FSD loss is also different from the CDS loss in Consistent3D, even when our view-dependent noise function gives the same noise for all camera views, implying an essential difference between our method and Consistent3D.

Table 5: Implementation details on 2D experiments with FSD.

| Methods Name | SDS | NFSD | VSD | FSD(ours) | DDIM |
|---|---|---|---|---|---|
| Iteration Num | 500 | 500 | 500 | 500 | 50 |
| CFG | 100 | 7.5 | 7.5 | 7.5 | 7.5 |
| Learning Rate | 2e-2 | 2e-2 | 2e-2 | 3e-3 | - |
| Optimizer | Adam | Adam | Adam | Adam | - |
| Timestep Annealing | linear | linear | linear | linear | - |

**Connection to Signal-ODE** Our reformulated ODE (Eq. 12) is equivalent to the Signal-ODE presented in the concurrent and independent work BOOT (Gu et al., 2023), which aims to distill a fast image generator. When the diffusion model is changed to sample prediction in Eq. 12, our reformulated PF-ODE is the same as the Signal-ODE in BOOT. In BOOT, they let the student image generation model predict the clean variables $\hat{x}_t^c$ on the ODE trajectory, while our method uses images rendered from 3D representation $\theta$ to model the clean variables $\hat{x}_t^c$ on the ODE trajectory.

## E  IMPLEMENTATION DETAILS

### E.1  3D EXPERIMENTS USING FSD

The backbone Diffusion Model for Fig. 1 in the main text is MVDream (Shi et al., 2023). The configuration for this figure is the same as Fig. 7 in the main text.

We use sqrt-annealing proposed by HiFA (Zhu & Zhuang, 2023) for 3D experiments in this work. We use the same CFG (Ho & Salimans, 2022) scale for SDS (Shi et al., 2023; Wang et al., 2023a) and FSD. We follow the default setting of threestudio (Guo et al., 2023) code base for other hyperparameters. We reimplemented ISM (Liang et al., 2023) in threestudio with the default parameter setting in ISM. We use the same number of iterations in 3D experiments. We mainly conduct our experiments using NeRF representation (Müller et al., 2022) since SDS-like methods are not sensitive to the form of 3D representations.

### E.2  2D EXPERIMENTS USING FSD

Here we describe the details of the experiments on 2D images with FSD in the main text. For Fig. 3 in the main text, we show the implementation details in Tab. 5. Below, we provide the loss functions for convenient reference.

Let denote the prediction of Diffusion Models as $\epsilon_y^t = \epsilon_\phi(x_t|y, t)$ and $c$ the classifier-free-guidance (CFG) (Ho & Salimans, 2022) scale. Then

$$\nabla_\theta L_{\text{SDS}}^\theta = \mathbb{E}_{\epsilon, c, t}\left[\left(c \cdot (\epsilon_y^t - \epsilon_\emptyset^t) + (\epsilon_\emptyset^t - \epsilon)\right) \frac{\partial g_\theta(c)}{\partial \theta}\right] \tag{17}$$

is the loss function of SDS (Poole et al., 2022; Wang et al., 2023a), where $\emptyset$ is the empty prompt.

$$\nabla_\theta L_{\text{NFSD}}^\theta = \mathbb{E}_{\epsilon, c, t}\left[\left(c \cdot (\epsilon_y^t - \epsilon_\emptyset^t) + (\epsilon_\emptyset^t - \epsilon_{y_{\text{neg}}}^t)\right) \frac{\partial g_\theta(c)}{\partial \theta}\right] \tag{18}$$

is the loss function of NFSD (Katzir et al., 2023), where $y_{\text{neg}}$ is a text negative prompt.

$$\nabla_\theta L_{\text{VSD}}^\theta = \mathbb{E}_{\epsilon, c, t}\left[\left(c \cdot (\epsilon_y^t - \epsilon_\emptyset^t) + (\epsilon_\emptyset^t - \epsilon_{\text{lora}}^t)\right) \frac{\partial g_\theta(c)}{\partial \theta}\right] \tag{19}$$

is the loss function of VSD (Wang et al., 2024), where $\epsilon_{\text{lora}}^t$ the Diffusion Model fine-tuned by LoRA (Hu et al., 2021).

$$\nabla_\theta L_{\text{FSD}}^\theta = \mathbb{E}_{c, t}\left[\left(c \cdot (\epsilon_y^t - \epsilon_\emptyset^t) + (\epsilon_\emptyset^t - \epsilon(c))\right) \frac{\partial g_\theta(c)}{\partial \theta}\right] \tag{20}$$

is the loss function of FSD. Additionally, $\boldsymbol{x}_t = \alpha_t \boldsymbol{g}_\theta(\boldsymbol{c}) + \sigma_t \boldsymbol{\epsilon}(\boldsymbol{c})$ for FSD.

Our re-implementation of $L_{NFSD}$ is slightly different from the original (Katzir et al., 2023) design of NFSD by ignoring the condition when $t < 200$, to keep the formulation simple. In this way, we find $\boldsymbol{\epsilon}_{\text{lora}}^t$ used in VSD (Wang et al., 2024) may work in a similar way as $\boldsymbol{\epsilon}_{y_{\text{neg}}}^t$ in NFSD (Katzir et al., 2023). As a result, single-particle VSD may not be able to generate diverse samples since it is still mode-seeking, aligning with the observation of ESD (Wang et al., 2023b). But we do not conduct further investigations which are out of the scope of our work.

## F  THEORY OF FLOW SCORE DISTILLATION

We will provide some additional preliminaries and proof of Proposition 1 in the main text in this section.

### F.1  DIFFUSION PF-ODE

The Diffusion PF-ODE (Song et al., 2020b) can be written in the following form:

$$\frac{\mathrm{d}\boldsymbol{x}_t}{\mathrm{d}t} = f(t)\boldsymbol{x}_t - \frac{1}{2}g^2(t)\nabla_{\boldsymbol{x}} \log p_t(\boldsymbol{x}_t|y), \quad \boldsymbol{x}_T \sim p_T(\boldsymbol{x}_T), \tag{21}$$

where $f(t) = \frac{\mathrm{d}\log\alpha_t}{\mathrm{d}t}$, $g^2(t) = \frac{\mathrm{d}\sigma_t^2}{\mathrm{d}t} - 2\frac{\mathrm{d}\log\alpha_t}{\mathrm{d}t}\sigma_t^2$, according to DPM-solver (Lu et al., 2022). To get Eq. (5) in the main text, we take the derivative of $\boldsymbol{x}_t/\alpha_t$:

$$
\begin{aligned}
\frac{\mathrm{d}(\boldsymbol{x}_t/\alpha_t)}{\mathrm{d}t} &= \frac{1}{\alpha_t}\frac{\mathrm{d}\boldsymbol{x}_t}{\mathrm{d}t} - \frac{\boldsymbol{x}_t}{\alpha_t^2}\frac{\mathrm{d}\alpha_t}{\mathrm{d}t} \\
&= \frac{1}{\alpha_t}\frac{\mathrm{d}\boldsymbol{x}_t}{\mathrm{d}t} - \frac{\boldsymbol{x}_t}{\alpha_t}f(t) \\
&= -\frac{g^2(t)}{2\alpha_t}\nabla_{\boldsymbol{x}} \log p_t(\boldsymbol{x}_t|y) \\
&= -\frac{\frac{\mathrm{d}\sigma_t^2}{\mathrm{d}t} - 2\frac{\mathrm{d}\log\alpha_t}{\mathrm{d}t}\sigma_t^2}{2\alpha_t}\nabla_{\boldsymbol{x}} \log p_t(\boldsymbol{x}_t|y) \\
&= -\left(\frac{1}{\alpha_t}\frac{\mathrm{d}\sigma_t}{\mathrm{d}t} - \frac{\sigma_t}{\alpha_t^2}\frac{\mathrm{d}\alpha_t}{\mathrm{d}t}\right)\sigma_t\nabla_{\boldsymbol{x}} \log p_t(\boldsymbol{x}_t|y) \\
&= \frac{\mathrm{d}(\sigma_t/\alpha_t)}{\mathrm{d}t}\left(-\sigma_t\nabla_{\boldsymbol{x}} \log p_t(\boldsymbol{x}_t|y)\right) \\
&= \frac{\mathrm{d}(\sigma_t/\alpha_t)}{\mathrm{d}t}\boldsymbol{\epsilon}_\phi(\boldsymbol{x}_t|t, y).
\end{aligned}
\tag{22}
$$

This equation is the scaled version of Diffusion PF-ODE under the notation of Karras et al. (2022).

### F.2  DDIM SAMPLING

We derive the DDIM (Song et al., 2020a) sampling algorithm in this section. According to the first order discretization of Eq. 22:

$$\frac{\boldsymbol{x}_t}{\alpha_t} - \frac{\boldsymbol{x}_s}{\alpha_s} = \left(\frac{\sigma_t}{\alpha_t} - \frac{\sigma_s}{\alpha_s}\right)\boldsymbol{\epsilon}_\phi(\boldsymbol{x}_s|s, y), \tag{23}$$

the sampling algorithm of DDIM (Song et al., 2020a) can be derived as the following equation:

$$\boldsymbol{x}_t = \alpha_t\left(\frac{\boldsymbol{x}_s - \sigma_s\boldsymbol{\epsilon}_\phi(x_s|y, s)}{\alpha_s}\right) + \sigma_t\boldsymbol{\epsilon}_\phi(\boldsymbol{x}_s|y, s). \tag{24}$$

### F.3  PROOF OF PROPOSITION 1

We present a detailed derivation of Proposition 1 in the main text in this section. We define

$$\boldsymbol{x}_t = \alpha_t\hat{\boldsymbol{x}}_t^c + \sigma_t\tilde{\boldsymbol{\epsilon}}, \tag{25}$$

for the reverse diffusion process and then we can apply change-of-variable on $\boldsymbol{x}_t$. Finally

$$
\begin{aligned}
\frac{\mathrm{d}\hat{\boldsymbol{x}}_t^{\mathrm{c}}}{\mathrm{d}t} &= \frac{\mathrm{d}\frac{\boldsymbol{x}_t - \sigma_t \tilde{\boldsymbol{\epsilon}}}{\alpha_t}}{\mathrm{d}t} \\
&= \frac{\mathrm{d}(\boldsymbol{x}_t/\alpha_t)}{\mathrm{d}t} - \frac{\mathrm{d}(\sigma_t/\alpha_t)}{\mathrm{d}t}\tilde{\boldsymbol{\epsilon}} \\
&= \frac{\mathrm{d}(\sigma_t/\alpha_t)}{\mathrm{d}t}(\boldsymbol{\epsilon}_\phi(\boldsymbol{x}_t|t, y) - \tilde{\boldsymbol{\epsilon}}).
\end{aligned}
$$

We also derive first-order discretization of FSD for 2D generation from Eq. 23:

$$
\hat{\boldsymbol{x}}_t^{\mathrm{c}} - \hat{\boldsymbol{x}}_s^o = (\frac{\sigma_t}{\alpha_t} - \frac{\sigma_s}{\alpha_s})(\boldsymbol{\epsilon}_\phi(\boldsymbol{x}_s|s, y) - \tilde{\boldsymbol{\epsilon}}), \tag{26}
$$

# G  VISUALIZE THE CHANGE-OF-VARIABLE

Even though we show FSD can generate similar images When using the same initial noise in the main text, there are notable differences between the generated images since FSD uses Adam (Kingma & Ba, 2014) to update parameters while DDIM uses first-order discretization of Diffusion PF-ODE. We show generation results of FSD that use first-order discretization in Fig. 16.

We also visualize the "*change-of-variable*" trick we used in the derivation of the main theorem. We define our new variable: the *clean image* $\hat{\boldsymbol{x}}_t^{\mathrm{c}}$ according to the following equation:

$$
\boldsymbol{x}_t = \alpha_t \hat{\boldsymbol{x}}_t^{\mathrm{c}} + \sigma_t \tilde{\boldsymbol{\epsilon}}. \tag{27}
$$

Notably, there is also another similar but different concept: the *one-step estimated ground-truth image* $\hat{\boldsymbol{x}}_t^{\mathrm{gt}}$, defined by:

$$
\boldsymbol{x}_t = \alpha_t \hat{\boldsymbol{x}}_t^{\mathrm{gt}} + \sigma_t \boldsymbol{\epsilon}_\phi(\boldsymbol{x}_t|y, t). \tag{28}
$$

We visualize the trajectory of $\boldsymbol{x}_t$, $\hat{\boldsymbol{x}}_t^{\mathrm{c}}$ and $\hat{\boldsymbol{x}}_t^{\mathrm{gt}}$ in the same DDIM generation process in Fig. 17. As $\hat{\boldsymbol{x}}_t^{\mathrm{gt}} - \hat{\boldsymbol{x}}_t^{\mathrm{c}} = \frac{\sigma_t}{\alpha_t}(\tilde{\boldsymbol{\epsilon}} - \boldsymbol{\epsilon}_\phi(\boldsymbol{x}_t|y, t)) \propto \nabla_\theta L_{\mathrm{FSD}}^\theta$ implies, we can see DDIM as a process that tries to align *clean image* with the *one-step estimated ground-truth image* generated by the Diffusion Model. We also visualize $\boldsymbol{x}_t$, $\hat{\boldsymbol{x}}_t^{\mathrm{c}}$ and $\hat{\boldsymbol{x}}_t^{\mathrm{gt}}$ of FSD and SDS in Fig. 18 and Fig. 19, respecitively.

# H  ALGORITHM FOR FLOW SCORE DISTILLATION

We provide a summarized algorithm for Flow Score Distillation in Algo. 1. Blender factor $\beta$ is set to 1 in all our experiments on FSD.

---
**Algorithm 1** Flow Score Distillation
---
1: **Input:** Text-to-image Diffusion Model $\boldsymbol{\epsilon}_\phi$ and prompt $y$. Learning rate $\eta$ for parameters of the 3D representation. A monotonically decreasing function $t(\tau)$. Blending factor $\beta$.
2: Compute $\boldsymbol{\epsilon}_b \sim \mathcal{N}(\boldsymbol{0}, \boldsymbol{I})$ and $\boldsymbol{\epsilon}_p \sim \mathcal{N}(\boldsymbol{0}, \boldsymbol{I})$.
3: **for** $\tau \in [0, \tau_{\mathrm{end}}]$ **do**
4:     Randomly sample camera parameter $\boldsymbol{c}$.
5:     Render image $\boldsymbol{g}_\theta(\boldsymbol{c})$ and opacity mask $\boldsymbol{M}$ from 3D representation $\theta$.
6:     Randomly sample $\boldsymbol{\epsilon} \sim \mathcal{N}(\boldsymbol{0}, \boldsymbol{I})$.
7:     $\boldsymbol{\epsilon}(\boldsymbol{c}) \leftarrow \sqrt{\beta} \cdot \left((1 - \boldsymbol{M}) \odot \boldsymbol{\epsilon}_b + \boldsymbol{M} \odot \boldsymbol{W}_{(W\frac{\phi_{\mathrm{cam}}}{2\pi}, H\frac{\theta_{\mathrm{cam}}}{\pi})}(\boldsymbol{\epsilon}_p)\right) + \sqrt{1 - \beta} \cdot \boldsymbol{\epsilon}$.
8:     $\theta \leftarrow \theta - \eta \cdot (\boldsymbol{\epsilon}_\phi(\boldsymbol{x}_t|y, t) - \boldsymbol{\epsilon}(\boldsymbol{c}))\frac{\partial \boldsymbol{g}_\theta(\boldsymbol{c})}{\partial \theta}$
9: **end for**
---

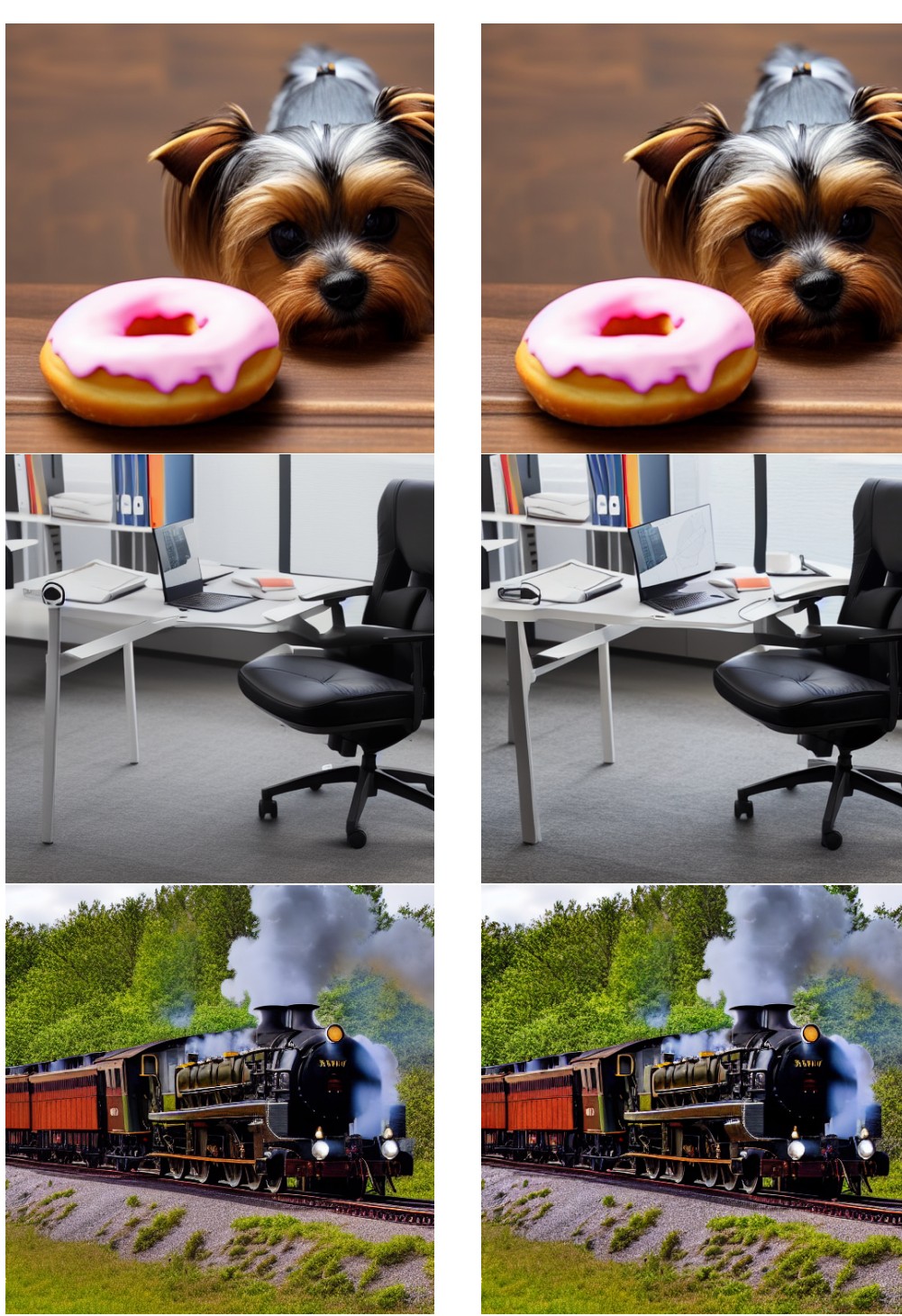

(a) FSD (first-order discretization)    (b) DDIM (Song et al., 2020a)

Figure 16: **Generation results of FSD and DDIM.** We apply FSD on 2D image generation using first-order discretization Eq. 26 instead of Adam (Kingma & Ba, 2014). In this case, we find FSD is the same as DDIM (Song et al., 2020a) except some negligible differences, which may come from the differences on handling initial conditions.

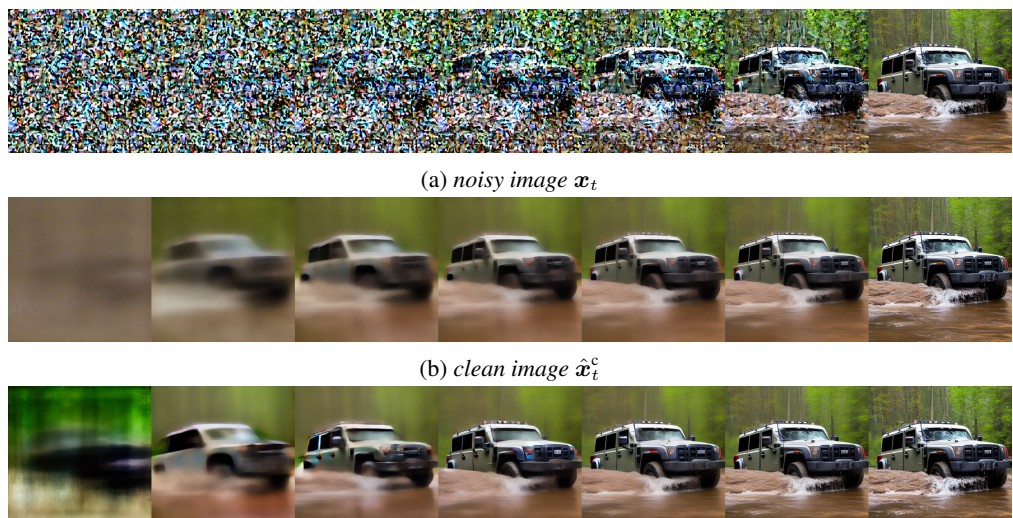

(a) *noisy image* $\boldsymbol{x}_t$

(b) *clean image* $\hat{\boldsymbol{x}}_t^{\mathrm{c}}$

(c) *one-step estimated ground-truth image* $\hat{\boldsymbol{x}}_t^{\mathrm{gt}}$

Figure 17: **Visualization of** $x_t$, $\hat{x}_t^{\mathbf{c}}$ **and** $\hat{x}_t^{\mathbf{gt}}$ **in DDIM generation process.** $\hat{\boldsymbol{x}}_t^{\mathrm{c}}$ is different from $\hat{\boldsymbol{x}}_t^{\mathrm{gt}}$. Moreover, one can see DDIM generation process as aligning $\hat{\boldsymbol{x}}_t^{\mathrm{c}}$ with $\hat{\boldsymbol{x}}_t^{\mathrm{gt}}$ since $\hat{\boldsymbol{x}}_t^{\mathrm{gt}} - \hat{\boldsymbol{x}}_t^{\mathrm{c}} \propto \nabla_\theta L_{\mathrm{FSD}}^\theta$

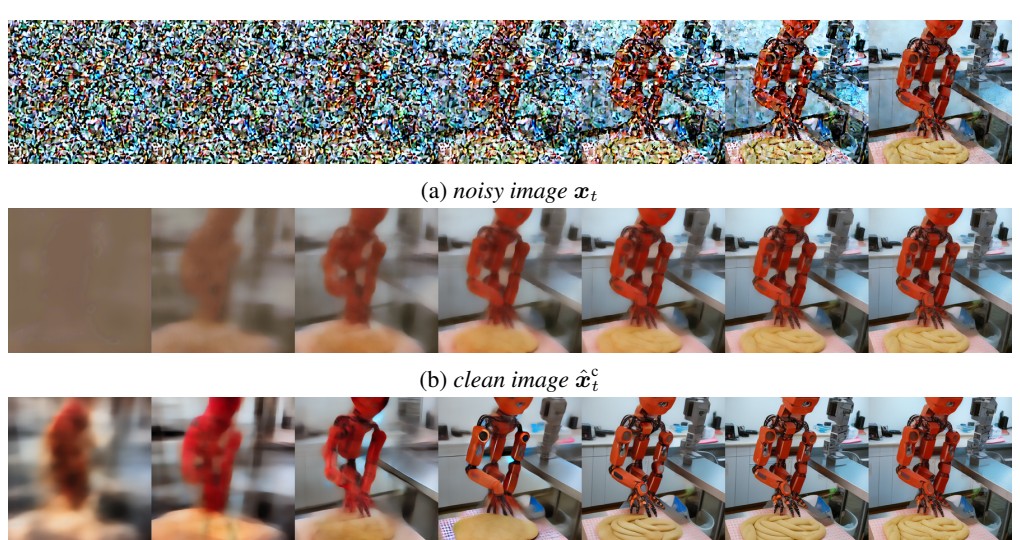

(a) *noisy image* $\boldsymbol{x}_t$

(b) *clean image* $\hat{\boldsymbol{x}}_t^{\mathrm{c}}$

(c) *one-step estimated ground-truth image* $\hat{\boldsymbol{x}}_t^{\mathrm{gt}}$

Figure 18: **Visualization of** $x_t$, $\hat{x}_t^{\mathbf{c}}$ **and** $\hat{x}_t^{\mathbf{gt}}$ **in FSD generation process.** Compared to DDIM, FSD uses Adam (Kingma & Ba, 2014) to update parameters. As a result, FSD needs more iterations to converge (See Tab. 5).

# I PRACTICAL DESIGNING RULES FOR $\tilde{\epsilon}$

## I.1 PRACTICAL DESIGNING RULES

We do not specify the form of $\epsilon(\boldsymbol{c})$ in the general form of FSD in the main text. However, it is intuitive to align $\epsilon(\boldsymbol{c})$ in 3D space like the noise priors used in Video Diffusion Models (Chang et al., 2024; Ge et al., 2023; Qiu et al., 2023). We have tried several designs of $\epsilon(\boldsymbol{c})$ and summarized several design rules for designing $\epsilon(\boldsymbol{c})$ as well as the related potential problems if violating the design rules for $\epsilon(\boldsymbol{c})$:

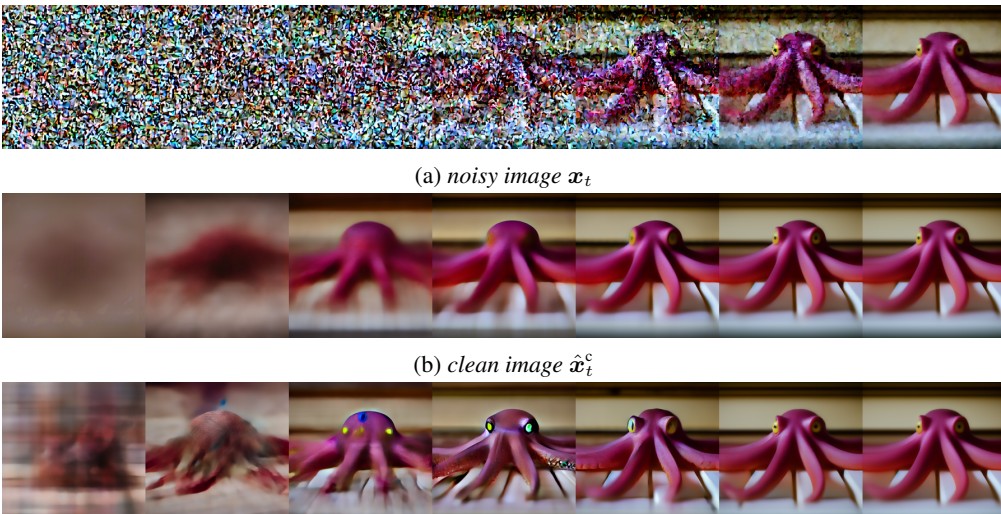

(a) *noisy image* $\boldsymbol{x}_t$

(b) *clean image* $\hat{\boldsymbol{x}}_t^{\mathrm{c}}$

(c) *one-step estimated ground-truth image* $\hat{\boldsymbol{x}}_t^{\mathrm{gt}}$

Figure 19: **Visualization of $\boldsymbol{x}_t$, $\hat{\boldsymbol{x}}_t^{\mathrm{c}}$ and $\hat{\boldsymbol{x}}_t^{\mathrm{gt}}$ in SDS generation process.** Compared to FSD, SDS adds random noise instead of fixed noise. As a result, the estimated GT images vary a lot and SDS needs a larger learning rate to converge (See Tab. 5).

- The noise associated with each camera view conditioning on camera view should be an uncorrelated Gaussian noise. i.e. $\boldsymbol{\epsilon}(\boldsymbol{c})|\boldsymbol{c} \sim \mathcal{N}(\boldsymbol{0}, \boldsymbol{I})$. If not so, the input image may be out-of-distribution for the pretrained text-to-image Diffusion Models.
- The noise associated with different camera views should be aligned in 3D space. Otherwise, FSD may degrade to the original SDS. This is consistent with the observation of recent work (Ge et al., 2023) on Video Generation, which showed that the noise maps corresponding to different video frames are highly correlated.
- The noise should not be only aligned in specific points in the 3D space. This may lead to broken geometry since the convergence speed in 3D space is not uniform (see analysis on failure of constant noise function in the main text).

### I.2 WORLD-MAP NOISE FUNCTION

We will take a deeper analysis of the property of the world map noise function in the main text. We will discuss how our proposed design rules are followed.

As noted in the main text, the local texture of the added noise is highly correlated with the generated images. In 3D spaces, it is natural to identify points, that correspond to similar points on the noise maps when projected onto the image planes of different camera views $\boldsymbol{c}$, as the points of high convergence speed. Let us denote those points as $\boldsymbol{p}_+$. For simplicity, we study the $\boldsymbol{p}_+$ that corresponds to the center points in image space.

For object-centric generation, a common camera view sampling strategy is to sample camera positions at $(r_{\mathrm{cam}}, \theta_{\mathrm{cam}}, \phi_{\mathrm{cam}})$ under spherical coordinate and force the camera to look at the center point. When we apply a constant noise function as $\boldsymbol{\epsilon}(\boldsymbol{c})$, the center point in 3D space is a $\boldsymbol{p}_+$ point. As a result, FSD trends to generate geometry with holes. When applying world-map noise function proposed in main text, $\boldsymbol{p}_+$ is located at $(r_+, \theta_{\mathrm{cam}}, \phi_{\mathrm{cam}})$. One can compute that

$$r_+^{\theta_{\mathrm{cam}}} \approx \frac{2 \tan \frac{\mathrm{FOV}}{2}}{2 \tan \frac{\mathrm{FOV}}{2} + \Theta} \cdot r_{\mathrm{cam}} \tag{29}$$

if consider nearby views with slightly different $\theta_{\mathrm{cam}}$ and

$$r_+^{\phi_{\mathrm{cam}}} \approx \frac{2 \tan \frac{\mathrm{FOV}}{2}}{2 \tan \frac{\mathrm{FOV}}{2} + \Theta \cdot \sin \theta_{\mathrm{cam}}} \cdot r_{\mathrm{cam}} \tag{30}$$

if consider views with slightly different $\phi_{\text{cam}}$. In this way, we align nearby views coarsely. Moreover, for different camera parameters, $r_+$ is different, avoiding nonuniform convergence speed in 3D space.

