# OpenReview forum: "Flow Score Distillation for Diverse Text-to-3D Generation"
_ICLR.cc/2025/Conference — ICLR 2025 Conference Withdrawn Submission_

### Official Review · Reviewer_FTbi · 2024-11-03

**Soundness:** 3
**Presentation:** 4
**Contribution:** 3
**Rating:** 5
**Confidence:** 5

**Summary:**

This paper aims to enhance SDS for generating diversity. It analyzes the sampling strategy of the SDS loss and establishes a connection between DDIM and the SDS generation process. The paper then proposes sampling deterministic noise and introduces a world-map noise function to align noise in 3D space. Additionally, it presents a coarse-to-fine pipeline to further address the Janus problem.

**Strengths:**

1. The paper is well-written and easy to understand.
2. The paper build a connection between SDS loss and the DDIM generation process, which is valuable and could inspire further research.
3. The proposed coarse-to-fine pipeline, though primarily an engineering solution, effectively contributes to generating high-quality 3D objects.

**Weaknesses:**

1. The concept of applying deterministic (fixed) noise to the SDS loss was introduced in previous work [1], but this prior work is neither cited nor discussed in the paper. My understanding is that the single-noise training in [1] is equivalent to the vanilla design described in Sec 4.1.1.


2. The proposed world-map noise function does not account for depth information -- noise is sampled without considering the shape of the generated object. Moreover, in the context of latent diffusion models, keeping the background noise static (as shown in Eq. 16) does not guarantee static gradients on pixels in the rendered image, since the gradient must pass through a non-linear, non-local VAE encoder. This issue is also noted in Chang et al. (2024) Appendix E. Therefore, it raises concerns about the technical soundness of this approach when applied to latent diffusion models.


[1] Diverse and Stable 2D Diffusion Guided Text to 3D Generation with Noise Recalibration, AAAI24

**Questions:**

see weaknesses

---

### Official Review · Reviewer_NWLZ · 2024-11-03

**Soundness:** 2
**Presentation:** 2
**Contribution:** 3
**Rating:** 5
**Confidence:** 3

**Summary:**

The paper focuses on improving the SDS (score-distillation sampling) method for 3D generation. Original SDS method optimizes the KL divergence between the data distribution and the distribution by the generated model, which lacks diversity. The paper first built a connection between the SDS loss and PF-ODE, and found that a consistent diffusion noise can be used for SDS. By using this method, the paper claimed that the diversity and quality of the 3D generation is improved.

**Strengths:**

- The paper built a connection between PF-ODE and SDS gradient to show that the SDS gradient is equivalence to some term in diffusion.
- The paper proposes to use a consistent noise (which is only possible with its FSD formulation) to encourage less multi-face problems.
- The paper shows that the proposed method can improve consistency.

**Weaknesses:**

- Inadequate analysis support to the claim of quality and diversity improvement. The paper claims that the proposing method has better quality and diversity comparing with previous method. I appreciate the FID analysis experiment. However, it is not convincing. Firstly, the FID is computed between the rendered images and generated images, where the rendered images are from 16 (prompt) x 4 (seed) = 64 3D objects (from my understanding). The amount of evaluated 3D objects is relatively lower. Secondly, the FID score is not reliable enough and it would be better to have a user study for the comparisons. In my mind, around 50 to 100 prompts for user study and 1K to 2K 3D objects for the metric evaluation would be better.

- The detailed process of FSD in Sec. 3.2 is not clear. I currently can only roughly construct its process from Table 1. It would be better to move some content in Appendix H (which includes a lot of other designs) to elaborate Sec 3.2. The paper also misses a connection between why using Adam to optimize Eqn 12 (a gradient to t instead of \theta) would lead to the generation goal.

- The paper would be better to make a clear separation between ground truth distribution, estimated distribution, and the auxiliary variables (usually from the definition). O.w., it would be hard for the reader to re-verify its correctness and leads to mis-concept.

   - In Eqn (5), I could accept an interchangeable use of $\epsilon_\phi$ and the gradient of $p_t$, but it would be better to have a clarification in the end of Sec 2.1.

    - It seems that Eqn (11) is a definition over $\hat{x}_t^c$ instead of $x_t$? The gradient of $x_t$  is directly used below in the derivation. From this context, I assume that it is the random variables "generated" from the original diffusion rule (i.e., the random variable with the forward diffusion distribution $p_t$). The current presentation is written in a way that $x_t$ is defined, which is a little bit confusing.

    - Following the above point, the \hat{x}_t^c in Eqn (11) and Eqn (8) are using the same notation but I think that their meaning should be different. In Eqn (8), it is the optimization parameters (an image or 3D objs). In Eqn (11), if it is also the optimization parameters, then we could not do gradient over $x_t$ any more.  Only once the $x_t$ is the distribution of the forward diffusion process at time step $t$, $x_t$ gradient is estimated by $\epsilon_\phi$. However, if $\hat{x}_t^c$ is the optimization parameters, this is not guaranteed.

Overall, I would trust the experimental results that the whole process should work. However, I would expect a revision over the math (especially the exact use of each derivation and each notation) and more proofs of the claimed improvements.

**Questions:**

- In Sec. 3.3, it's better to have some visualization to illustrate the "SDS are over-smoothness and lacks diversity" issue instead of just describing by text. This part of analysis is less convincing to the reader but I roughly get the idea.

- What would be the loss function to optimize for the 2D example in Sec. 3.2? Would that be the same to the objective introduced in Sec. 4?

---

### Official Review · Reviewer_Rzc5 · 2024-11-09

**Soundness:** 2
**Presentation:** 3
**Contribution:** 1
**Rating:** 3
**Confidence:** 3

**Summary:**

This paper comes up with a simple noise-sampling strategy to address diversity limitations in Score Distillation Sampling (SDS) for text-to-3D generation. It leverages world-map noise function that enables more varied 3D outputs by aligning noise coherently across different camera views and a two-stage generation pipeline.

**Strengths:**

- This paper formulates SDS as a generalized DDIM (Denoise Diffusion Implicit Models) process and introduces a world-map noise function for 3D generation, the noise mechanism design is simple and seems effective.
- I like the quality of its generated 3D assets, which are sharp and come with fine-grained details, the results quality is consistent across various examples from main paper and supp.
- it also reveals the relationships between initial noise map to final 3D assets, which is insightful

**Weaknesses:**

- While I like the quality of plotted 3D assets examples, the biggest concern I think is on the contribution significance and novelty. All the strategies like leveraging multi-view diffusion models, scheduled noise level annealing, SDS formulation are already extensively explored in previous works starting from MV-Dream, ProlificDreamer, VSD. The innovation on noise map sampling is kind weak
- I think the comparison to baselines are also not fair and informative enough, as the proposed method is using MVDream as a stronger pre-trained 2D diffusion model
- FSD improves over other methods but still requires significant time for generation, raising questions about practical deployment for real-time 3D generation applications. Other recent text-to-image-to-3D can already generate high-quality 3D assets in seconds
- While the world-map noise function is introduced, further clarification is needed on how this function consistently maintains diversity across different views without affecting 3D consistency.

**Questions:**

NA

---

### Official Review · Reviewer_8djh · 2024-11-10

**Soundness:** 2
**Presentation:** 1
**Contribution:** 1
**Rating:** 3
**Confidence:** 4

**Summary:**

This paper proposes a Flow Score Distillation (FSD) method to combine Score Distillation Sampling (SDS) with the proposed noise sampling strategy. After explaining the connection between DDIM and SDS, this paper suggests using the same noise throughout the optimization process in SDS. For text-to-3D generation, the authors use a world map noise function, which renders noise given specific camera parameters after initializing a 3D-shape of noise. The results show that the proposed technique generates varied 3D objects with different noise initializations.

**Strengths:**

S1. This paper aims to solve an important problem: the lack of diversity in generated 3D results by score distillation.

S2. The proposed world-map noise function is interesting.

S3. The results show varied 3D objects given the same prompt without performance degradation compared with SDS.

**Weaknesses:**

W1. Lack of novelty and originality. The existing papers have already discussed the connection between DDIM and SDS [NewRef-1, NewRef-2] and using a fixed noise for SDS [NewRef-2, NewRef-3]. In addition, the paper does not include enough rationales of how the proposed noise sampling can resolve the issues in using a fixed noise.


W2. Insufficient experiments. The paper lacks in-depth analysis on the proposed noise sampling technique. In addition, some results of previous methods show much difference from the results in the original papers.


W3. Presentation. I think some notations are confusing or used without definition. For example, $\hat{\mathbf{x_{t}^{c}}}$ in Line 234 is not related to $t$. Section 4.1.2 uses many undefined variables such as $D, H, W, H_\text{hidden}, W_\text{hidden}$. $\mathbf{W}_{(\cdot)}(\epsilon_p)$ is not an operation since it refers to a noise patch. In addition, the operation $\mathbf{W}$ is not defined in the paper, although it is the most important part of the proposed method.



[NewRef-1] Kim et al., “DreamSampler: Unifying Diffusion Sampling and Score Distillation for Image Manipulation”, ECCV 2024.

[NewRef-2] Lukoianov et al., “Score Distillation via Reparametrized DDIM”, NeurIPS 2024.

[NewRef-3] Alldieck et al., Score Distillation Sampling with Learned Manifold Corrective, ECCV 2024.

**Questions:**

Q1. Regarding the connection between DDIM and SDS, how different is the analysis in this paper from [NewRef-1]?

Q2. I think Figure 14 is insufficient to claim the limitation of using a fixed noise.
- Can the authors provide more examples with different prompts?
- Is there any case that the proposed noise sampling also makes the flaws?
- Can a simple alternative such as using multiple fixed noises resolve the issue, rather than the proposed method?
- Can the statements in Section 4.1.1 be analyzed empirically and validated enough?


Q3. How is the $\mathbf{W}$ operation defined?

Q4. Why is the background/foreground noise defined separately? Would simply using the world-map noise function for a fixed noise for 3D be insufficient?

Q5. As Figure 4, can the author provide the results of generated images via the world-map noise function? If the assumption on the world-map noise function holds, I think the generated images should show 3D-consistent changes like the results in Figure 4.

Q6. I think the quality of existing methods in this paper seems to be much degraded compared with the results in the original papers (e.g. NFSD and VSD). Can the authors explain the reasons?

---

### Note · Authors · 2024-11-13

I have read and agree with the venue's withdrawal policy on behalf of myself and my co-authors.